# Measuring statistics-induced entanglement entropy with a Hong–Ou–Mandel interferometer

Gu Zhang[1,2,5], Changki Hong [3,5], Tomer Alkalay[3,5], Vladimir Umansky[3], Moty Heiblum [3] ✉, Igor Gornyi [2] ✉ & Yuval Gefen [4] ✉

Despite its ubiquity in quantum computation and quantum information, a universally applicable definition of quantum entanglement remains elusive. The challenge is further accentuated when entanglement is associated with other key themes, e.g., quantum interference and quantum statistics. Here, we introduce two novel motifs that characterize the interplay of entanglement and quantum statistics: an 'entanglement pointer' and a 'statistics-induced entanglement entropy'. The two provide a quantitative description of the statistics-induced entanglement: (i) they are finite only in the presence of quantum entanglement underlined by quantum statistics and (ii) their explicit form depends on the quantum statistics of the particles (e.g., fermions, bosons, and anyons). We have experimentally implemented these ideas by employing an electronic Hong–Ou–Mandel interferometer fed by two highly diluted electron beams in an integer quantum Hall platform. Performing measurements of auto-correlation and cross-correlation of current fluctuations of the scattered beams (following 'collisions'), we quantify the statistics-induced entanglement by experimentally accessing the entanglement pointer and the statistics-induced entanglement entropy. Our theoretical and experimental approaches pave the way to study entanglement in various correlated platforms, e.g., those involving anyonic Abelian and non-Abelian states.

A pillar of quantum mechanics—quantum entanglement—prevents us from obtaining a full independent knowledge of subsystem $A$ entangled with another subsystem $B$. Indeed, the state of subsystem $A$ may be influenced or even determined following a measurement of $B$, even when both are distant apart. This feature, known as the non-locality of quantum entanglement, is at the heart of the fast-developing field of quantum information processing (see, e.g., refs. 1–5). An apt example is a system comprising two particles with opposite internal magnetic moments (spin up and spin down). Imagine we put one particle on Earth (subsystem $A$) and the other on Mars (subsystem $B$). If

measurement on $A$ reveals the particle is in the up state, this instantaneously dictates that the $B$ particle is down. Following Bell[6] and CHSH[7] inequalities, measurements of the respective spins in different directions may unambiguously demonstrate the quantum nature of the entanglement of $A$ and $B$.

An essential way in which quantum entanglement reveals itself is the entangled subsystem's entropy. The entanglement entropy (EE) of subsystem $A$ can be found when the complete information of $B$ is discarded. This amounts to summing over all possible states of $B$. Formally, the von Neumann EE is defined as $S_{ent} = -\,\mathrm{Tr}\,(\rho_A \ln \rho_A)$,

[1]Beijing Academy of Quantum Information Sciences, Beijing, China. [2]Institute for Quantum Materials and Technologies, Karlsruhe Institute of Technology, Karlsruhe, Germany. [3]Braun Center for Submicron Research, Department of Condensed Matter Physics, Weizmann Institute of Science, Rehovot, Israel. [4]Department of Condensed Matter Physics, Weizmann Institute of Science, Rehovot, Israel. [5]These authors contributed equally: Gu Zhang, Changki Hong, Tomer Alkalay. ✉e-mail: moty.heiblum@weizmann.ac.il; igor.gornyi@kit.edu; yuval.gefen@weizmann.ac.il

where $\rho_A = \mathrm{Tr}_B(\rho_{AB})$ is the reduced density matrix of $A$ after tracing over the states of $B$, where $\rho_{AB}$ is the density matrix in the entire Hilbert space $\mathcal{H}_A \otimes \mathcal{H}_B$. When the subsystems share common entangled pairs of particles, such pairs are effectively counted by the EE.

Another pillar of quantum mechanics is the quantum statistics of indistinguishable particles, whose wavefunction might acquire a non-trivial phase upon exchanging the particles' positions (braiding). This phase is pertinent in classifying quasiparticles as fermions, bosons, and, most interestingly, anyons. Being instrumental in realizing platforms for quantum information processing (see, e.g., ref. 8), it motivated several insightful experiments[9–15] that intended to detect anyonic statistics[16–23]. Among such experimental setups, the Hong–Ou–Mandel (HOM) interferometer[24] was employed as one of the simplest platforms to manifest bosonic[25], fermionic (see, e.g., refs. 26,27), and anyonic (Laughlin quasiparticles)[14,21,28] statistics. Despite their importance, the interplay of entanglement with quantum statistics has hardly been studied, either theoretically or experimentally (see, however, refs. 27,29).

In an attempt to address EE in the context of quantum transport, it has been theoretically proposed[30] to focus on a single quantum point contact (QPC) geometry (with partitioning $\mathcal{T}$ of the incident beam), which allows partial separation of two subsystems (arms), $A$ and $B$. Following partial tracing over states in one subsystem, the EE can, in principle, be obtained indirectly via a weighted summation over even cumulants of particle numbers extracted from the current-noise measurements (see the discussion of noise cumulants in, e.g., ref. 31). However, even measurement of the fourth cumulant is not straightforward[32] in mesoscopic conductors[33]. To our knowledge, no study of EE through measurements of quantum transport has been reported. We note that the EE had been measured in localized atomic systems (see ref. 34 for a review). In addition, the impurity entropy (not an EE), induced by frustration at quantum criticality was most recently reported in refs. 35,36.

In the present study, we fuse two foundational quantum-mechanical notions: quantum statistics and entanglement, and propose the concept of statistics-induced entanglement. We introduce two functions quantifying entanglement arising from the quantum statistics of indistinguishable particles: (i) the entanglement pointer (EP, $\mathcal{P}_E$) and (ii) the statistics-induced entanglement entropy (SEE) (denoted as $S_{SEE}$). Both are derived from correlations of current fluctuations in an HOM configuration and are expressed in Eqs. (1) and (4). Importantly, these two functions vanish for distinguishable particles, and are finite when indistinguishable particles are emitted from the two sources $\mathcal{S}_A$ and $\mathcal{S}_B$ become entangled following collisions.

Typically, entanglement is the outcome of Coulomb interaction between distinct constituents of the system. Here, we focus on the entanglement being solely a manifestation of quantum statistics. If one considers current–current correlators, this contribution to the entanglement may be complemented (or even fully masked) by the effect of Coulomb interactions between the colliding particles. Below we show, theoretically and experimentally, that with our specially designed function, $S_{SEE}$, the leading contributions of the Coulomb interaction are canceled, with the remaining terms dominated by quantum statistics [cf. Eq. (3) and Eq. (S21) of Supplementary Information (SI) Sect. S1].

We note that the acquisition of statistics-induced entanglement is both instantaneous and non-local. It is acquired immediately when two identical particles braid each other, even at a distance. By these features it is universal. By contrast, the Coulomb interaction contribution to the entanglement requires the two particles to directly interact with each other, and depends on the strength and duration of this interaction, hence it is non-universal. This non-universal influence from interaction becomes dominant in the measured noise (see below), but is negligible in our constructed EP, $\mathcal{P}_E$.

Turning now to the technicalities of our study, the theoretical derivation of the explicit forms of the EP and SEE (see "Methods") employs, respectively, the Keldysh technique[31] and an extended version of the approach of ref. 30 [see Eq. (8)]. The actual measurements were carried out in a HOM configuration[24], fabricated in a two-dimensional electron gas (2DEG) tuned to the integer quantum Hall (IQH) regime. Two highly diluted (via weak partitioning in two outer QPCs) edge modes were let to collide at a center-QPC, and current fluctuations (shot noise) of two scattered diluted beams (Fig. 1) were measured. While the definitions of the EP and SEE are not restricted to a specific range of parameters, expressing SEE in terms of the measured EP is possible only within the limit of highly diluted impinging current beams [Eq. (7)]. As will be shown, the theoretical prediction agrees very well with the experimental data.

## Results
### The model and the EP
Our HOM interferometer consists of four arms, all in the IQH regime (Fig. 1). Two sources $\mathcal{S}_A$ and $\mathcal{S}_B$ are biased equally at $V_A = V_B = V_{\mathrm{bias}}$, with sources currents weakly scattered by two QPCs, each with dilution $\mathcal{T}_A$ and $\mathcal{T}_B$, respectively. The partitioned beams impinge on a central QPC (from middle arms $\mathcal{M}_A$ and $\mathcal{M}_B$ in Fig. 1) with transmission $\mathcal{T}$[21]. The two transmitted electron beams are measured at drains $\mathcal{D}_A$ and $\mathcal{D}_B$.

With this setup, we define the first entanglement-quantification function—EP, $\mathcal{P}_E$. It is expressed through the cross-correlation of the two current fluctuations, excluding the statistics-irrelevant contribution,

$$
\begin{aligned}
\mathcal{P}_E(\mathcal{T}_A, \mathcal{T}_B, V_{\mathrm{bias}}) &\equiv \int dt \Big[ \langle I_A(t) I_B(0) \rangle_{\mathrm{irr}}|_{\mathcal{T}_A, \mathcal{T}_B, V_{\mathrm{bias}}} \\
&- \langle I_A(t) I_B(0) \rangle_{\mathrm{irr}}|_{0, \mathcal{T}_B, V_{\mathrm{bias}}} - \langle I_A(t) I_B(0) \rangle_{\mathrm{irr}}|_{\mathcal{T}_A, 0, V_{\mathrm{bias}}} \Big].
\end{aligned} \tag{1}
$$

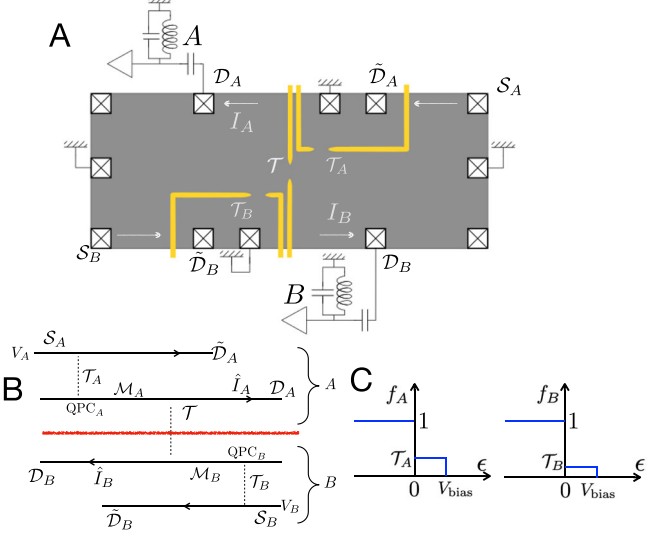

**Fig. 1 | Schematics of the setup. A** Schematics of the experimental setup. **B** The corresponding theoretical schematics. The Hong–Ou–Mandel interferometer consists of two sources ($\mathcal{S}_A$ and $\mathcal{S}_B$) and two diluted middle arms ($\mathcal{M}_A$, $\mathcal{M}_B$) via transmission probabilities $\mathcal{T}_A$ and $\mathcal{T}_B$ (with two quantum point contacts). The currents are measured at drains $\mathcal{D}_A$ and $\mathcal{D}_B$. For later convenience, we call the source arms after the diluters as $\tilde{\mathcal{D}}_A$ and $\tilde{\mathcal{D}}_B$. For simplicity, the two sources are equally biased: $V_A = V_B = V_{\mathrm{bias}}$. The red line separates the two entangled subsystems, $A$ (the two upper arms with labels $A$) and $B$ (the two lower arms with labels $B$). **C** Distribution functions of the two middle arms ($\mathcal{M}_A$ and $\mathcal{M}_B$) for non-interacting fermions ($f_A$, $f_B$, respectively) at zero temperature (carrying shot noise). The double-step distributions are modified when the filling factor is larger than one, with an added interaction between the two modes on each edge [Eq. (S34) of Supplementary Information Sect. S2].

Here, $I_A$ and $I_B$ refer to the current operators in the corresponding drains $\mathcal{D}_A$ and $\mathcal{D}_B$ (Fig. 1B), and "irr" refers to the irreducible correlators (connected correlation function), where the product of the averages is removed. Note that the last two terms in Eq. (1) are each evaluated with only one active source (i.e., either $\mathcal{T}_A$ or $\mathcal{T}_B$ is zero), and, thus do not involve the two-particle scattering in the HOM configuration[19,26,37]. This removal of last two terms has been carried out in refs. 37,38, however without referring to entanglement. Importantly, Eq. (1) yields zero for distinguishable, non-interacting particles, since the first term is then a superposition of two independent single-source terms. By contrast, the EP is finite and statistics-dependent for indistinguishable particles.

For instance, for a double-step-like distribution of such particles (e.g., Fig. 1C for fermions), we obtain cross-correlations (CC) of current operators,

$$\text{fermions: } \int dt \langle I_A(t)I_B(0)\rangle_{\text{irr}}|_{\mathcal{T}_A,\mathcal{T}_B,V_{\text{bias}}}$$
$$= -\frac{e^3}{h}\mathcal{T}(1-\mathcal{T})\left[(\mathcal{T}_A-\mathcal{T}_B)^2 + \mathcal{T}_A\mathcal{T}_B P_{\text{QPC}}\right]V_{\text{bias}},$$
$$\text{bosons: } \int dt \langle I_A(t)I_B(0)\rangle_{\text{irr}}|_{\mathcal{T}_A,\mathcal{T}_B,V_{\text{bias}}} \qquad (2)$$
$$= \frac{e^3}{h}\mathcal{T}(1-\mathcal{T})\left[(\mathcal{T}_A-\mathcal{T}_B)^2 - \mathcal{T}_A\mathcal{T}_B P_{\text{QPC}}\right]V_{\text{bias}}.$$

Here $P_{\text{QPC}}$ describes an additional bunching (or anti-bunching) probability induced by Coulomb interactions within the central QPC (cf. SI Sect. S2). Note that for equal diluters, $\mathcal{T}_A = \mathcal{T}_B$, the non-interacting part of the CC vanishes, indicating that the nature of the CC is then solely determined by interactions. This is however not so for the EP. Indeed, with Eqs. (1) and (2), we obtain,

$$\text{fermion EP: } \mathcal{P}_E = (2 - P_{\text{QPC}})\frac{e^3}{h}\mathcal{T}(1-\mathcal{T})\mathcal{T}_A\mathcal{T}_B V_{\text{bias}},$$
$$\text{boson EP: } \mathcal{P}_E = (-2 - P_{\text{QPC}})\frac{e^3}{h}\mathcal{T}(1-\mathcal{T})\mathcal{T}_A\mathcal{T}_B V_{\text{bias}}. \qquad (3)$$

In the presence of a weak inter-mode interaction among particles within the middle arms $\mathcal{M}_A$ and $\mathcal{M}_B$, $P_{\text{QPC}}$ is replaced with $P_{\text{QPC}} + P_{\text{frac}}$ [see "Methods" and Eq. (S51) of SI Sect. S2]. The term $P_{\text{frac}}$ refers to the influence of intra-arm charge fractionalization that produces particle-hole dipoles in the two interacting edge modes (refs. 39,40). Crucially, the unavoidable Coulomb-interaction contribution to the EP, parameterized by $P_{\text{QPC}}$ and $P_{\text{frac}}$ (introduced in Supplementary Eqs. (S50) and (S51), respectively), appears in terms that are quadratic in the beam dilution ($\mathcal{T}_A$ and $\mathcal{T}_B$) and hence is parametrically smaller than the linear ($\sim \mathcal{T}_A P_A$ and $\sim \mathcal{T}_B P_B$) terms in the noise correlation functions [see Eq. (13) in "Methods"]. It follows that the EP rids of the undesired effect of Coulomb interactions, hence truly reflecting the state's statistical nature.

## EE from statistics

The second entanglement quantifier is the SEE, which is defined in a similar spirit to the EP (by removing the statistics-irrelevant single-source contributions to the EE),

$$S_{\text{SEE}}(\mathcal{T}_A,\mathcal{T}_B) \equiv -\left[S_{\text{ent}}(\mathcal{T}_A,\mathcal{T}_B) - S_{\text{ent}}(\mathcal{T}_A,0) - S_{\text{ent}}(0,\mathcal{T}_B)\right]. \qquad (4)$$

To illustrate the relation between the SEE and Bell-pair entanglement, we consider the case of two incoming fermions (Fig. 2A). The pure two-particle state at the output of our device is represented as (see Fig. 2)

$$|\Psi\rangle = |\tilde{\psi}\rangle + |\psi\rangle. \qquad (5)$$

Here, $|\tilde{\psi}\rangle \equiv |\psi_{2,0}\rangle + |\psi_{0,2}\rangle$ denotes a state where both particles end up in either subsystem $A$ (2,0) or $B$ (0,2) (Fig. 2B, C, respectively).

In either case, obeying Pauli's blockade, two electrons must occupy the two arms of the same subsystem: for example, $\tilde{\mathcal{D}}_A$ and $\mathcal{D}_A$ for the state (2,0), which can be written as (1,1,0,0) in the basis of drain arms $\tilde{\mathcal{D}}_A, \mathcal{D}_A, \mathcal{D}_B, \tilde{\mathcal{D}}_B$. Any coupling between the arms within one subsystem cannot change the 1,1 arrangement for subsystem $A$ in $|\psi_{2,0}\rangle$. This implies that no quantum manipulations on subsystem $A$, leading to Bell's inequalities (refs. 6,7) are possible with $|\tilde{\psi}\rangle$ alone, i.e., without coupling the present setup to extra channels. The same holds for subsystem $B$. Nevertheless, $|\tilde{\psi}\rangle$ is a non-product state with nonlocal[41,42] entanglement: if the two particles are detected in subsystem $A$, this automatically implies that no particles are to be detected in subsystem $B$. In principle, Bell's inequalities can be tested with $|\tilde{\psi}\rangle$ using modified devices akin to those proposed for a similar bosonic state (a NOON state) in, e.g., refs. 43,44 and refs. 45,46, after the introduction of external states.

By contrast,

$$|\psi\rangle = \alpha|\uparrow_A\rangle|\uparrow_B\rangle + \beta|\downarrow_A\rangle|\downarrow_B\rangle \equiv |\psi_{1,1}\rangle, \qquad (6)$$

represents an effective Bell pair (with amplitudes $\alpha$ and $\beta$), where one particle leaves the device through subsystem $A$ and the other through $B$ (hence $|\psi_{1,1}\rangle$, as opposed to $|\psi_{2,0}\rangle$ and $|\psi_{0,2}\rangle$, see Fig. 2D, E). Here, $|\uparrow_A\rangle, |\downarrow_A\rangle$ are certain (mutually orthogonal) linear combinations of the states $|\tilde{\mathcal{D}}_A\rangle$ and $|\mathcal{D}_A\rangle$; $|\uparrow_B\rangle, |\downarrow_B\rangle$ are defined similarly for subsystem $B$ [see Eq. (S67) of SI Sect. S3]. The quantum superposition in Eq. (6) allows for rotating the pseudospin (i.e., rotation between orthonormal bases $|\uparrow_A\rangle$ and $|\downarrow_A\rangle$) of subsystem $A$, hence the measurement of pseudospins in a transverse direction is possible, as required by Bell's inequalities. It is also worth noting that the states $|\uparrow_A\rangle, |\downarrow_A\rangle$ are non-local: each of them is constructed out of states in the arms associated with $\tilde{\mathcal{D}}_A$ and $\mathcal{D}_A$, which are spatially separated by the middle arm $\mathcal{M}_A$ of length about $2\,\mu$m in the real setup, see below. The detailed forms of $|\tilde{\psi}\rangle$ and $|\psi\rangle$ are manifestations of quantum statistics, which underlines the statistics-induced entanglement, captured by the function $S_{\text{SEE}}$. Motivated by the conditions of low temperature and minimal noise underlining the experimental platform, we have focused here on the ideal case of pure states. Departing from these conditions, one needs to consider mixed states, in which case probing Bell's nonlocality is more demanding than addressing quantum entanglement[42] (cf. Sect. S7 of SI).

Technically, the building blocks of SEE, $S_{\text{ent}}$ [Eq. (4)] can be obtained[30,47,48] by calculating the generating function $\chi(\lambda) \equiv \sum_q \exp(i\lambda q)P_q$ of the full counting statistics (FCS)[31] (see "Methods"). Here $P_q$ refers to the probability to transport charge $q$ between the subsystems $A$ and $B$ over the measurement time. In a steady state, $S_{\text{SEE}}$ is proportional to the dwell time $\tau$ (see SI Sect. S5), which corresponds to the shortest travel time from the central QPC to an external drain. This should be replaced by the coherence time, $\tau_\varphi$, if the latter is shorter than the dwell time. The EE grows linearly with the coherent arm's length (beyond this coherence length the particles become dephased, hence disentangled). In the following analysis, we neglect externally induced dephasing along the arms, as the dephasing length in this type of IQH device is known to be longer than the arm's length.

While the function $S_{\text{SEE}}$ is in principle measurable, it becomes readily accessible in the strongly diluted limit $\mathcal{T}_A, \mathcal{T}_B \ll 1$. In this limit, $S_{\text{SEE}}$ is approximately equal to a function $\tilde{S}_{\text{SEE}}$, which is proportional to the EP of free fermions and bosons (see "Methods"):

$$S_{\text{SEE}} \xrightarrow{\mathcal{T}_A,\mathcal{T}_B \ll 1} -\frac{1}{2e^2}\mathcal{P}_E \ln(\mathcal{T}_A\mathcal{T}_B\mathcal{T}^2)\tau \equiv \tilde{S}_{\text{SEE}}. \qquad (7)$$

Note the statistics-sensitive factor, $\mathcal{P}_E$. Since the non-interacting (purely statistical) contribution to $\mathcal{P}_E$ [cf. Eq. (3)] for bosons is opposite in sign to its fermionic counterpart, so is the corresponding $\tilde{S}_{\text{SEE}}$.

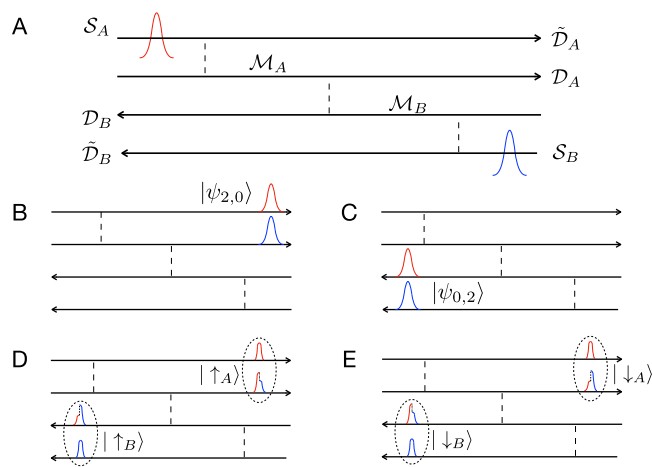

**Fig. 2 | Two components $|\bar{\psi}\rangle$ and $|\psi\rangle$, of two-particle wavefunctions, shown in the schematics of Fig. 1B. A** Pre-collision configurations. Two particles (red and blue pulses) are injected from $\mathcal{S}_A$ and $\mathcal{S}_B$, respectively. Hereafter, post-scattering quasi-particles comprise contributions from both incident particles (blue and red). **B, C** Constituents of the state $|\bar{\psi}\rangle = |\psi_{2,0}\rangle + |\psi_{0,2}\rangle$. **D, E** Constituents of the Bell pair state $|\psi\rangle = \alpha|\uparrow_A\rangle|\uparrow_B\rangle + \beta|\downarrow_A\rangle|\downarrow_B\rangle$, Eq. (6). In comparison to $|\bar{\psi}\rangle$ configurations, particle states of $|\psi\rangle$ configurations are entangled, both within (indicated by dashed ellipses) and between (indicated by the red–blue mixed pulses) subsystems.

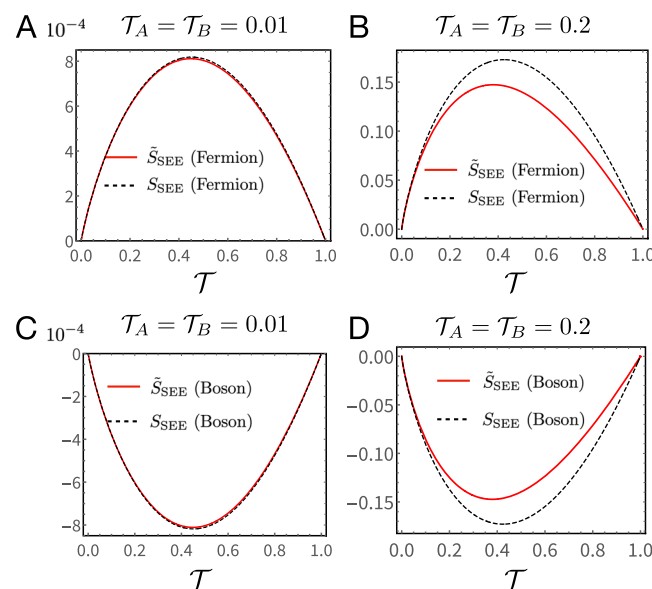

**Fig. 3 | Comparison between theoretical values of $S_{SEE}$ and $\widetilde{S}_{SEE}$.** Free fermions (**A, B**) and free bosons (**C, D**). These functions perfectly overlap for the entire range of $\mathcal{T}$ when $\mathcal{T}_A = \mathcal{T}_B = 0.01$ in (**A, C**). When $\mathcal{T}_A = \mathcal{T}_B = 0.2$ (**B, D**), a finite but small difference begins to show up between them. The bias used $V_{bias} = 20.7\,\mu V$. We take the dwell time $\tau_{dwell} = 0.01$ ns (see Supplementary Information Sect. S5 for the evaluation of $\tau_{dwell}$).

Importantly, addressing SEE (compared to $S_{ent}$) has two obvious upsides. First, extracting $\mathcal{P}_E$ via current cross-correlation measurements, this quantity is easy to obtain in the strongly diluted particle beam limit. Second, it allows us to rid of most of the undesired effects of Coulomb interactions, and clearly single out quantum statistics contributions. To validate Eq. (7) we compare $S_{SEE}$ [calculated according to Eq. (S21) of SI Sect. S1] and $\tilde{S}_{SEE}$ for free fermions (Fig. 3A, B) and free bosons (Fig. 3C, D). We next compare our theoretical predictions with experiments.

**Experimental results**
The experimental structure was fabricated in uniformly doped GaAs/AlGaAs heterostructure, with an electron density of $9.2 \times 10^{10}$ cm$^{-2}$ and 4.2 K dark mobility $3.9 \times 10^{6}$ cm$^2$ V$^{-1}$ s$^{-1}$. The 2DEG is located 125 nm below the surface. Measurements were conducted at an electron temperature -14 mK. The structure is shown in Fig. 1A (schematically) and in Fig. 4A (electron micrograph). Two QPCs are used to dilute the two electron beams, which collided at the central QPC located 2 μm away. Two amplifiers, each with an LC circuit tuned to 730 KHz (with bandwidth 44 KHz) measuring the charge fluctuations, are placed at a large distance (around 100 μm) from the 2D Hall bar. The outer-most edge mode of filling factor $v = 3$ of the IQH was diluted by the two external QPCs.

Cross-correlation of the current fluctuations of the reflected diluted beams from the central QPC ($\mathcal{T} = 0.53$), with $\mathcal{T}_A = \mathcal{T}_B = 0.2$, is plotted in Fig. 4B. The corresponding single source CC, with $\mathcal{T}_A = 0.2$ and $\mathcal{T}_B = 0$, is plotted in Fig. 4C. Though the data is rather scattered, the agreement with the theoretically expected CC is reasonable. For both cases, the measured data displays a clear deviation from the non-interacting curve: evidence of strong interaction influence. Importantly, for the equal-source situation ($\mathcal{T}_A = \mathcal{T}_B = 0.2$, Fig. 4B), the CC is entirely produced by interactions within a single source [see SI Eq. (S48)], indicating the inadequacy of CC to quantify entanglement.

The measured data, with the applied source voltage larger than the electron's temperature ($eV > k_B T$) was used to calculate the EP (Fig. 4D) and the SEE (Fig. 4E), and then compared with the expected EP and SEE. The measurement results conformed with the theoretical

prediction of the EP. More data for $v = 3$ and $v = 1$ situations is provided in SI Sects. S8 and S9, respectively.

As discussed above, the current fluctuations are also influenced by two sources of Coulomb interactions: (i) inter-mode interaction at the same edge, and (ii) interaction within the central QPC in the process of the two-particle scattering (Fig. 4B, C). However, the influence of these interactions on the EP and SEE is negligible in our setup, which is in great contrast to, e.g., refs. 49,50, where entanglement is purely interaction-induced; see also refs. 51,52, as Fig. 4D, E demonstrate (see also SI Sect. S2). Thus, the measured current noise indeed yields information on statistics-induced entanglement.

## Discussion and outlook
Entanglement and exchange statistics are two cornerstones of the quantum realm. Swapping quantum particles affects the many-body wavefunction by introducing a statistical phase, even if the particles do not interact directly. We have shown that quantum statistics induces genuine entanglement of indistinguishable particles, and developed theoretical and experimental tools to unambiguously quantify this effect. Our success in ridding of the contribution of the local Coulomb interaction, facilitates a manifestation of the foundational property of quantum mechanics−nonlocality. It also presents the prospects of generalizing our protocol to a broad range of correlated systems, including those hosting anyons (Abelian and non-Abelian, see e.g., SI Sect. S7), and exotic composite particles (e.g., neutralons at the edge of topological insulators[53,54]). Our protocol may also be generalized to include setups based on more complex edge structures, and platforms where the quasi-particles involved are spinful. Another intriguing direction is to explore the interplay of statistic-induced entanglement and quantum interference (e.g., similar structures considered in SI Sect. S8).

## Methods
In Methods, we provide (i) the sample fabrication, (ii) the experimental setups, (iii) an outline of the SEE derivation, (iv) a physical

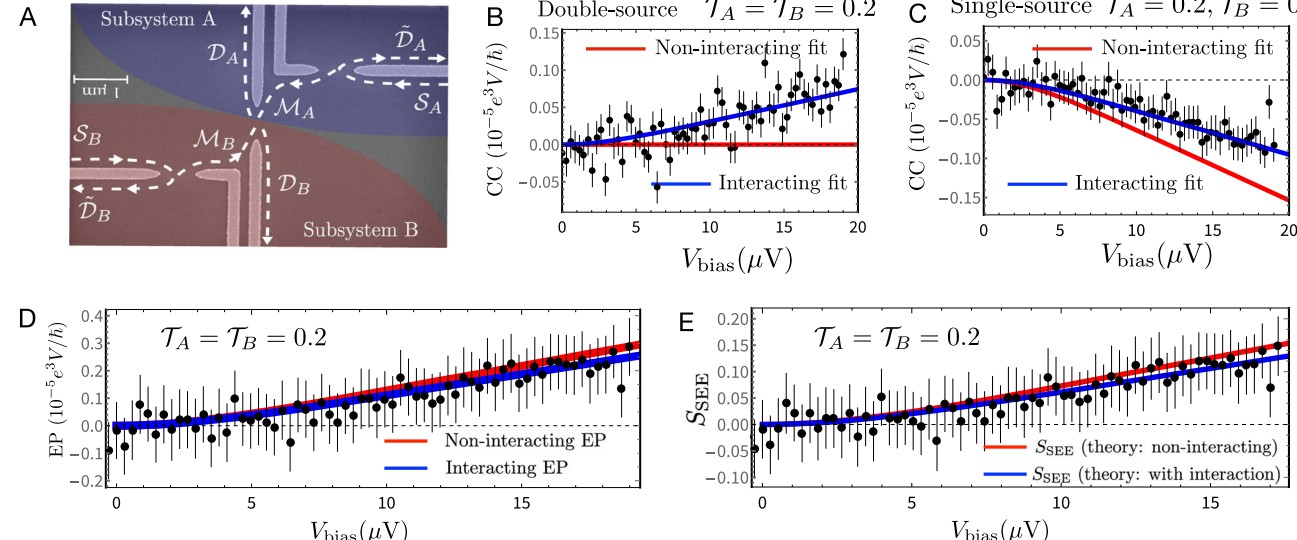

**Fig. 4 | Experimental setup and experiment-theory comparisons. A** Scanning electron microscope micrograph of the central part of the fabricated sample. Subsystems $A$ and $B$ (cf. Fig. 1B) are highlighted by shaded blue and shaded red areas, respectively. Transport directions of edge states in the arms associated with the sources $\mathcal{S}_{A,B}$ and drains $\mathcal{D}_{A,B}$, $\tilde{\mathcal{D}}_{A,B}$, as well as in the diluted middle arms $\mathcal{M}_{A,B}$, are indicated by dashed black arrows. **B, C** Double-source and single-source cross-correlations (CCs), respectively [see Eq. (S48) of the Supplementary Information (SI) for expressions that include interaction contributions]. In both cases, theoretical curves with interaction taken into account (blue curves) agree better with the experimental data (black dots). **D, E** Measured data for the entanglement pointer (EP) and statistics-induced entanglement entropy (SEE). **D** Compares the measured EP (black dots) with theoretical curves: including interaction contributions (blue) and non-interacting particles (red). Although the interaction strength is the same as

in (**B, C**), the difference between the theoretically calculated values of the EP with and without interaction contribution is much smaller than for the cross-current correlations, in panels (**B, C**). This demonstrates that the expression for the EP subtracts the interaction contribution to the leading order. **E** Comparison between the experimentally measured data (black dots) and the theoretical dependence of $S_{\mathrm{SEE}}$ on the source current (here $\tau_{\mathrm{dwell}} = 0.01$ ns as in Fig. 4). Experimental data points are obtained in two steps (see SI Sect. S1D for details): (i) we evaluate $\bar{S}_{\mathrm{SEE}}$ (using Eq. (7)) with the measured EP from panel (**D**), and, (ii) relying on the fact that at the experimental value $\mathcal{T} = 0.53$ the ratio $S_{\mathrm{SEE}}/\bar{S}_{\mathrm{SEE}} \approx 1.22$ in (**B**), we use this ratio to scale the measured $\bar{S}_{\mathrm{SEE}}$ to reconstruct $S_{\mathrm{SEE}}$. Both the interacting (blue) and non-interacting (red) theoretical curves for $S_{\mathrm{SEE}}$ agree remarkably well with the experimental data. The error bars represent the standard deviation of the mean for the set of measurements (see SI for more details).

interpretation of the EP-SEE connection, and (v) a physical picture of the effect of interaction along the arms: charge fractionalization.

## Sample fabrication
The GaAs/AlGaAs heterostructure we use consists of a 2DEG layer (with 125 nm depth) and a donor layer (with 92 nm depth). We perform the MESA layer (with a thickness of 100 nm) etching with a wet solution $H_3PO_4:H_2O_2:H_2O = 1:1:50$. Ge/Ni/Au materials (with a stacked total thickness of 450 nm) serve as a rapid thermal annealed standard Ohmic contacts. A high-$\kappa$ oxide layer ($HfO_2$, thickness 30 nm) is fabricated with an atomic layer deposition machine. Ti/Au QPC gates with 200 nm width and an 800 nm gap, are deposited on the oxide layer. Before the deposition of the last 300 nm Ti/Au metal contact pad, the sample is etched for reactive-ion etching with Ar and $BCl_3$ gas window for the oxide layer for the contact.

## Experimental setups
The base temperature for the measurement was below 10 mK by the CMN temperature sensor and Rutinum Oxide sensor. A low frequency of 13 Hz was used for the transmission and gate voltage versus conductance measurement by lock-in amplifier (NF corporation LI 5655). For the noise measurement, a voltage source with a 1 GΩ resistor series is connected for the DC currents to the source ohmic contacts. Each amplifier line used around ~730 KHz resonance frequency LC resonant circuit with blocking capacitor and amplified by ATF-34143 HEMT based home-made voltage preamplifier at 4 K plate. At room temperature, the noise amplified the room temperature voltage amplifier (NF corporation SA-220F5) and measured the noise

with digital multi-meters (HP 34401A) after a homemade analog cross-correlator.

## Derivation of SEE
The system EE $S_{\mathrm{ent}}$ (and the ensued SEE) can be obtained through its connection with FCS[30,47,48], i.e.,

$$S_{\mathrm{ent}} = \frac{1}{\pi} \int_{-\infty}^{\infty} du \frac{u}{\cosh^2 u} \, \mathrm{Im} \left[ \ln \chi(\pi - \eta - 2iu) \right], \tag{8}$$

where $\chi$ refers to the generating function that fully describes the tunneling between subsystems, and $\eta$ is a positive infinitesimal.

Of the non-interacting situation, the zero-temperature generating function equals[31,55],

$$\ln \chi(\lambda_{\mathrm{FCS}}) = eV_{\mathrm{bias}}\tau \ln\Big\{1 + \mathcal{T}\big[\mathcal{T}_A(1 - \mathcal{T}_B)\big(e^{i\lambda_{\mathrm{FCS}}} - 1\big)\big] \\ + \mathcal{T}\big[\mathcal{T}_B(1 - \mathcal{T}_A)\big(e^{-i\lambda_{\mathrm{FCS}}} - 1\big)\big]\Big\}/h, \tag{9}$$

where $\tau$ is the dwell time and $\lambda_{\mathrm{FCS}}$ is an auxiliary field (the measuring field) introduced in FCS. To the second order of dilutions, it approximately becomes

$$\frac{S_{\mathrm{ent}}(\mathcal{T}_A, \mathcal{T}_B)}{\tau} \approx \frac{eV_{\mathrm{bias}}}{h}\Big\{\mathcal{T}\big[\mathcal{T}_A + \mathcal{T}_B - \mathcal{T}_A \ln(\mathcal{T}_A\mathcal{T}) \\ - \mathcal{T}_B \ln(\mathcal{T}_B\mathcal{T})\big] - \frac{1}{2}\mathcal{T}^2\big(\mathcal{T}_A^2 + \mathcal{T}_B^2\big) \\ + \mathcal{T}(1 - \mathcal{T})\mathcal{T}_A\mathcal{T}_B \ln\big(\mathcal{T}^2\mathcal{T}_A\mathcal{T}_B\big) + O\big(\mathcal{T}_{A,B}^3\big)\Big\}. \tag{10}$$

After the removal of the single-source contributions, Eq. (10) reduces to Eq. (7), i.e.,

$$
\begin{aligned}
S_{\text{SEE}} &= -[S_{\text{ent}}(\mathcal{T}_A, \mathcal{T}_B) - S_{\text{ent}}(\mathcal{T}_A, 0) - S_{\text{ent}}(0, \mathcal{T}_B)] \\
&\approx -\frac{eV_{\text{bias}}}{h}\mathcal{T}(1-\mathcal{T})\mathcal{T}_A\mathcal{T}_B \ln(\mathcal{T}^2\mathcal{T}_A\mathcal{T}_B).
\end{aligned}
\tag{11}
$$

The non-perturbative SEE expressions can be found in Eqs. (S21) of SI Sect. S1 and (S60) of SI Sect. S2.

## Understanding the relation between EP and SEE

To understand the EP–SEE relation Eq. (7), we start with a single-particle situation where one particle from source $\mathcal{S}_A$ contributes to the entanglement

$$
\begin{aligned}
S_{1e} &= \mathcal{T}_A\mathcal{T}\ln(\mathcal{T}_A\mathcal{T}) + (1-\mathcal{T}_A)\ln(1-\mathcal{T}_A\mathcal{T}) \\
&+ \mathcal{T}_A(1-\mathcal{T})\ln(1-\mathcal{T}_A\mathcal{T})
\end{aligned}
$$

that has three contributions. The second one $(1-\mathcal{T}_A)\ln(1-\mathcal{T}_A\mathcal{T})$ comes from the case when the particle stays in the upper source $\mathcal{S}_A \to \tilde{\mathcal{D}}_A$. This term has no contribution to SEE as the involved particle has no chance to join a two-particle scattering. Of the remaining two, the first one $\mathcal{T}_A\mathcal{T}\ln(\mathcal{T}_A\mathcal{T})$ dominates in the strongly diluted regime. It can be considered as the product of the conditioned EE $\ln(\mathcal{T}_A\mathcal{T})$ and its corresponding probability $\mathcal{T}_A\mathcal{T}$.

Now we move to two-particle scatterings to see the role of statistics. Following the single-particle analysis, a two-particle scattering event produces the leading conditioned EE $[\ln(\mathcal{T}\mathcal{T}_A) + \ln(\mathcal{T}\mathcal{T}_B)]$ when two particles enter the same drain (i.e., bunching) after the scattering. Consequently, for a two-particle scattering event of free particles, the difference in the EE emerges between indistinguishable (fermions and bosons) and distinguishable cases,

$$
\begin{aligned}
\text{fermion:} \quad &(P^b_{\text{fermion}} - P^b_{\text{dis}})[\ln(\mathcal{T}\mathcal{T}_A) + \ln(\mathcal{T}\mathcal{T}_B)]/2 \\
&= -\mathcal{T}(1-\mathcal{T})[\ln(\mathcal{T}\mathcal{T}_A) + \ln(\mathcal{T}\mathcal{T}_B)], \\
\text{boson:} \quad &(P^b_{\text{boson}} - P^b_{\text{dis}})[\ln(\mathcal{T}\mathcal{T}_A) + \ln(\mathcal{T}\mathcal{T}_B)]/2 \\
&= \mathcal{T}(1-\mathcal{T})[\ln(\mathcal{T}\mathcal{T}_A) + \ln(\mathcal{T}\mathcal{T}_B)],
\end{aligned}
\tag{12}
$$

where $P^b_{\text{fermion}}$, $P^b_{\text{boson}}$, and $P^b_{\text{dis}}$ refer to the bunching probabilities of fermions, bosons, and distinguishable particles, respectively. In more realistic considerations, the two-particle scattering rate, i.e., $\mathcal{T}_A\mathcal{T}_B$ should be included as the prefactor of Eq. (12), leading to the statistics-induced entropy of 2e-scattering processes $-\mathcal{P}_E[\ln(\mathcal{T}\mathcal{T}_A) + \ln(\mathcal{T}\mathcal{T}_B)]/2$ for fermions and bosons alike. It equals the auxiliary function $\tilde{S}_{\text{SEE}}$ [Eq. (7)] after the integral over energy and time.

Notice that the arguments above on the EP-SEE connection rely on the fact that both quantities can describe the tunneling between two subsystems. As a consequence, this EP-SEE connection remains valid if EP is instead defined after replacing $I_A$ and $I_B$ of Eq. (1) by the total current of two subsystems $A$ and $B$. This flexibility in the definition of EP enhances the potential range of applicability of our theory.

## Effect of interaction along the arms: charge fractionalization

In the main text, we mention that interaction along the arms influences correlation functions, EP, and SEE via the introduction of charge fractionalization[39,40]. To understand this phenomenon, we consider two chiral fermionic channels 1 and 2. Both channels (with corresponding fields $\phi_1$ and $\phi_2$) are described by free 1D Hamiltonians

$$
H_1 = \frac{v_F}{4\pi\hbar}\int dx(\partial_x\phi_1)^2, \quad H_2 = \frac{v_F}{4\pi\hbar}\int dx(\partial_x\phi_2)^2.
$$

Fermions in channels also Coulomb-interact, leading to

$$
H_{12} = \frac{v}{2\pi\hbar}\int dx(\partial_x\phi_1)(\partial_x\phi_2).
$$

The total Hamiltonian can be diagonalized via the rotation $\phi_\pm \equiv (\phi_1 \pm \phi_2)/\sqrt{2}$, after which two modes $\phi_\pm$ travel at different velocities $v_F \pm v$. Consequently, after entering one middle arm, an electron gradually splits into two (spatially separated) wave packets. With fractionalization taken into consideration, the cross current–current correlation becomes (see SI Sect. S2)

$$
\begin{aligned}
\int dt\langle I_A(t)I_B(0)\rangle_{\text{irr}} = &-\frac{e^3}{h}\mathcal{T}(1-\mathcal{T})\Big[(\mathcal{T}_A - \mathcal{T}_B)^2 \\
&+ \mathcal{T}_A P_A + \mathcal{T}_B P_B + \mathcal{T}_A\mathcal{T}_B(P_{\text{QPC}} + P_{\text{frac}})\Big]V_{\text{bias}},
\end{aligned}
\tag{13}
$$

where

$$
\begin{aligned}
P_A &= -\left[(1-\mathcal{T}_A)\left(1 - \frac{1}{2-2\mathcal{T}}\right) + \frac{l^2}{\lambda^2}\right]\frac{v^2}{v_F^2}, \\
P_B &= -\left[(1-\mathcal{T}_B)\left(1 - \frac{1}{2-2\mathcal{T}}\right) + \frac{l^2}{\lambda^2}\right]\frac{v^2}{v_F^2},
\end{aligned}
\tag{14}
$$

refer to the modification of correlation function due to the particle fractionalization in each arm, and

$$
P_{\text{frac}} = 2\frac{v^2}{v_F^2}\frac{l^2}{\lambda^2},
\tag{15}
$$

only contributes when both sources are on. In these expressions, $l$ and $\lambda$ refer to the distance from the diluter to the central QPC, and the half-width of the diluted fermionic wave packet, respectively. Following the equations above, the extent of fractionalization in our system depends on the interaction amplitude $v$ and the distance $l$ from the diluter to the central QPC (around 2 μm in our setup). Based on experimental data (cf. SI Sect. S2), we expect the fractionalization to be minimal before the packets arrive at the central QPC.

## Data availability

The data used in this study are available in the figshare database [https://doi.org/10.6084/m9.figshare.23546235].

## Code availability

We do not develop code for this work.

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

## Acknowledgements

We acknowledge Dong E. Liu, Gabriele Campagnano, Christian Glattli, and Janine Splettstoesser for useful comments, and Yunchul Chung and

Hyungkook Choi for the discussion of the experiments. I.G. and Y.G. acknowledge the support from the DFG grant No. MI658/10-2 and German-Israeli Foundation (GIF) grant No. I-1505-303.10/2019. Y.G. acknowledges support from the Helmholtz International Fellow Award, the DFG Grant RO 2247/11-1, CRC 183 (project C01), the US-Israel Binational Science Foundation, and the Minerva Foundation. M.H. acknowledges the continuous support of the Sub-Micron Center staff and the support of the European Research Council under the European Union's Horizon 2020 research and innovation programme (grant agreement number 833078).

## Author contributions

G.Z., I.G., and Y.G. conducted all theoretical calculations and analysis. C.H. and T.A. fabricated the structures, performed measurements, and analyzed the data. V.U. designed and grew the heterostructures by molecular-beam epitaxy. M.H. supervised the experiments. All the authors participated in writing the manuscript.

## Funding

## Competing interests

The authors declare no competing interests.
