## [Peer Review File · Nature Communications]

Measuring statistics-induced entanglement entropy with a Hong-Ou-Mandel interferometerReviewer #1 (Remarks to the Author):

The ability to measure entanglement entropy (EE) in condensed matter systems would have important implications on experiments testing proposals for the realization of exotic electronic states, where theory often finds that the best metric to characterize such states is via EE.

While the authors of this paper claim to have demonstrated this ability, in my reading I don't see the fundamental difference between the measurements proposed and then carried out in this work, compared to earlier noise measurements probing exchange statistics in the same system. For example, I don't understand:

- how the metrics EP and SEE relate to entanglement as I perceive it from, e.g., thinking about Bell's inequalities and two-particle spin or polarization states
- how EP and SEE are related to, or distinct from, previous measurements of exchange statistics in quantum Hall "collider" setups such as this one
- how current noise measurements at DA and DB implement quantum measurements of the type that would be required to test Bell's inequalities.

Overall I would say this: this paper may be an important one for the fraction of physicists who are very close to this field. But for readers who are not working on precisely this topic, either theoretically or experimentally, the impact will be lost: it is not clear how it moves us beyond what was possible before. What experiment or analysis could the authors perform that was not possible before, and how would the answer be important? I feel like the answer may be hiding somewhere in the discussion of Fig 4, but after multiple times rereading this section I don't see it clearly.

Reviewer #2 (Remarks to the Author):

Dear authors and editors,

with much interest I have read the manuscript "Measuring statistics-induced entanglement entropy with a Hong-Ou-Mandel interferometer". Below, I provide detailed feedback.

Summary of the manuscript

The authors investigate the entanglement content of (quasi-)particles in two spatial regions separated by a quantum point contact (QPC) with respect to their quantum statistics, theoretically as well as experimentally. Their considered setup comprises a two-dimensional electron gas on top of which two diluted electron beams interfere via an electronic analog of the Hong-Ou-Mandel (HOM) interferometer. The entanglement generated by the QPC geometry is accessed in terms of two observables, that is, the entanglement pointer (EP) and the statistics-induced entanglement entropy (SEE), which are designed to capture the quantum statistical contribution to the entanglement of the underlying quasi-particles by subtracting single-source contributions. For instance, the SEE differs by an overall sign for bosons and fermions, see Fig. 3. How this approach is related to Bell non-locality, how to incorporate the effects of additional Coulomb interactions and how the two entanglement quantifiers can be extracted experimentally in the limit of strong dilution is thoroughly discussed. Additionally, the authors realize their proposal and provide evidence for the utility of their methods by comparing how much the interacting and non-interacting curves differ for the cross-correlation function with respect to the EP as well as the SEE. Detailed calculations, additional experimental evidence and an outlook for other applications of their methods is discussed in a supplementary file.

Originality and significance

I consider the main results, i.e. the EP/SEE and their experimental extraction, interesting and significant. The originality of the EP/SEE is certainly debatable: as acknowledged by the authors, the EP has been discussed in Ref. [37]. Further, the overall idea of subtracting a certain contribution one wants to get rid of, which underpins both observables of interest, is rather straightforward. Nevertheless, the finding that the SEE becomes directly accessible in the limit of strong dilution is strong as it is backed theoretically and experimentally beyond the non-interacting case, i.e. with Coulomb interactions being present. With their method, the authors solve a crucial problem in the area of quantum transport: determining the quantum statistical contribution to the entanglement of quasi-particles. While this alone may not merit publication in a high-impact journal such as Nature Communications, it is definitely a solid basis for it to be considered.

Technical quality and scientific

Although my overall impression is that the work is technically sound (I was particularly happy with the comprehensive supplement), I have major as well as minor concerns regarding the scientific rigor and several questions.

1. I was very surprised that the authors did neither comment on the systematic nor on the statistical errors in their experimental data analysis for Figs. 4B-E, S5 and S6 (has the experiment actually been repeated for a given V_{bias} , i.e. is there any information regarding statistics?). I consider this information crucial especially for evaluating Figs. D and E in the regime of large V_{bias} . It seems that the interacting curve fits better, but without error bars this statement is weak. More generally, it is always important to state the experimental uncertainty somewhere, otherwise the data points may be overinterpreted and the overall significance may be weakened.

2. The authors nicely show that the SEE is accessible in the limit of strong dilution and compare the approximation underlying these findings to higher dilutions in Fig. 3. However, the experimental curves of the SEE in Fig. 4 were extracted beyond the strong dilution regime and computed by a simple rescaling of the strong diluted curves from Fig. 3, which is explained in a bit more detail in S1D. Unfortunately, this method is nowhere justified rigorously, presumably because the two types of curves are obviously not related by a rescaling (this is clearly evident from Figs. 3 B and D). I ask the authors to either give a mathematical rigor justification for why a rescaling is appropriate (and not other simple operations such as an addition) at $T_A=T_B=0.2$ and $T=0.53$ or to provide any other substantial argument.

3. The manuscript lacks scientific rigor whenever concepts of quantum information are discussed in more detail. First, the important distinction between 'Bell non-locality' and 'entanglement' is not made (see e.g. PRA 104, 052208 (2021) for a recent study I came across when writing this report). In simple words, Bell non-locality is a particularly strong form of entanglement, and detecting true Bell non-locality (for which witnessing entanglement does not suffice), is a notoriously difficult task (which the authors should also mention in their introduction and find appropriate experimental references for).

The confusion of Bell non-locality (see Brunner et al. for a review, which should definitely be cited somewhere) with entanglement becomes more severe in the subsection 'Entanglement entropy from statistics' and the corresponding SM section S3. The authors have chosen to describe the state ψ_C as 'classically entangled', which I consider imprecise and misleading (the classical counterpart to

entanglement is simply mixedness, which is rather weak form of correlations). The state ψ_C does actually correspond to the Hong-Ou-Mandel state, a particular kind of a NOON state, which has been investigated in detail in the literature [to be consistent with the literature on this state the authors should also write $\psi_C = \psi_{2,0} - \psi_{0,2}$ and define $\psi_{0,2}$ accordingly in the SM, which actually comes with a minus sign by default, see (S63)]. Interestingly, this state is not only entangled, but also non-local in the sense that it allows for EPR steering (the hierarchy of entanglement is: Bell non-local > EPR steering > entanglement > separable), see PRA 94, 042119 (2016). Hence, manipulations on one subsystem can influence the other subsystems, but the quantum correlations are not strong enough for Bell non-locality. In contrast, the state ψ_Q , which does actually correspond to the Bell state (the authors should also mention that), exhibits Bell non-locality and was the state used in the first study demonstrating the violation of Bell's inequality.

To summarize, the paragraph below (5) and the corresponding discussion in the SM have to be rewritten, the distinction between 'quantum' and 'classical' entanglement should be changed to 'Bell non-locality' and 'entanglement' throughout the manuscript and the states ψ_C and ψ_Q should be described in more detail, including references to the literature.

4. Based on the latter considerations, I assume that SEE is actually a quantifier for Bell non-locality. If this is true, this would be a very strong result and I encourage the authors to investigate this deeper. In any case, while I find the comparison with the Bell pairs nice and instructive, I'm not convinced that the SEE does indeed only capture Bell non-locality. It might well be that it also evaluates to a non-zero real number if the state under considerations exhibits EPR steering. Hence, I ask the authors to provide a mathematical rigor argument for why SEE is zero if and only if the state exhibits Bell non-locality. If such a statement can not be provided, a few statement in the paragraph below Eq. (6) have to be weakened.

Also, I ask the authors to clarify whether the SEE is a true entanglement measure, an entanglement monotone, or an entanglement witness (the term 'entanglement quantifier' used by the authors is rarely used and usually means entanglement monotone, but I have also seen it as a synonym for entanglement witness), see also my comment 5 below.

Out of curiosity: the definition of the SEE, ie. Eq. (4), looks very similar to the definition of the mutual information. I'm not sure, but may the two are related. I encourage the authors to briefly investigate their relations, but this is optional. My motivation is the following: if there is a simple relation, it may become obvious why the SEE does only capture Bell non-locality.

5. In S6 the authors claim that the SEE is able to quantify the entanglement of globally mixed states. This is usually the field of entanglement measures, of which the authors have mentioned and referenced two prominent ones. Proving that these two are indeed entanglement measures is not straightforward at all, and I ask the authors to support their claim with mathematics or instead weaken their statements.

6. Out of interest: the theoretical discussion comprises both fermions and bosons, while the experimental discussion does (of course) not. Is there an easy way of investigating the entanglement of bosons with a similar setup, e.g. by considering Cooper-pairs?

7. Also out of interest: in the discussion the authors state that their method can be generalized to anyons. However, the measurable SEE relies on the entanglement pointer, which itself does only distinguish bosons/fermions by the overall sign and hence, there is no room left. Can you provide any intuition for why your ansatz does not break down for anyons?

The answers to the latter two questions can be very brief.

Presentation

Overall, the manuscript is written well and sufficiently accessible and self-contained. Nevertheless, a few statements were confusing to me and I also found some typos and inconsistencies.

1. The definition of the entanglement entropy in the introduction is wrong. It lacks a crucial overall minus sign [the definition in the SM is correct in (S10), but also wrong above (61)]. Maybe this also implies that the overall minus sign in Eq. (4) should be dropped, but this is anyway mere convention.
2. The usage of bra's and ket's to denote state vectors is inconsistent throughout the manuscript. For instance, in the paragraph below Eq. (6) the state vectors up/down are written without ket's. Or, in (S58), the authors have equated a real number with a state vector in Hilbert space.
3. Similarly, the usage of operator hats is inconsistent. The current operators are equipped with operator hats, while all other operators are not. Here, a single clarifying sentence should suffice.
4. In the information theory literature, it is rather uncommon to denote the composition of two subsystems (or random variables) A and B by $A + B$. The latter usually refers to the sum of the two, while their composition is denoted by AB . Note also that the entropy of the sum is a lower bound on the entropy of the composition. Further, measures of correlations between A and B are usually written with $A:B$, see e.g. in the caption of Fig. S6.
5. It is 'von Neumann' and not 'Von Neumann', see e.g. the sentence above (S61).
6. What the authors call 'irreducible correlation' is usually referred to as 'connected correlation function' and written with a subindex c.
7. Some abbreviations are not defined, are defined after their first usage or are defined multiple times. For example, 'SEM' is not defined, '2DEG' is defined too late, and 'HOM' is defined twice.
8. There is maybe an obsolete space bar in the last down ket w.r.t. B in the caption of Fig. 2.
9. In the majority of the manuscript, the authors use "", but sometimes they instead use " (see e.g. 'collisions' or 'generating' function).
10. Below Eq. (6) the authors write ""pseudospin" of subsystem A'. Please elaborate briefly what is meant here.
11. I recommend the authors to simplify the comparison of Figs. S5 and S6 to the main Fig. 4 by showing the quantities of interest as functions of V_{bias} (if there is no good reason against it).

Recommendation

Altogether, the results are of high significance, especially because of their experimental accessibility, and the manuscript is written in an accessible way, which warrants its publication in a high-impact journal with a broad scope. However, the manuscript in its current form lacks rigor regarding mainly, first, central concepts of entanglement and quantum information theory and second, the extraction process of the SEE. I'm convinced that a major revision following my comments would substantially strengthen the results and would put them into a proper perspective in the lore of entanglement theory. When reading through the manuscript, several follow-up questions regarding information theoretic aspects came to my mind, and I believe that the authors would benefit a lot from providing a

thorough discussion of Bell non-locality etc. for the reader.

To cut a long story short, given that the authors can satisfyingly respond to my comments, I can warmly recommend publication in Nature Communications!

Optional suggestions

Additionally to my previous comments, I encourage the authors to consider a few comments regarding subtleties. Note, however, that I do not expect the authors to abide to any of the following suggestions. Hence, whether the authors implement and/or reply to my comments is optional.

1. The authors have chosen to denote the system's total state by ρ_0 . Given that ρ_1 to ρ_4 label eigenvalues of the reduced density matrix, see e.g. (S61), I suggest to use ρ or ρ_{AB} instead.

2. I personally would find it more convenient to have Figs. 1 (A) and (B) in the same order, that is, having subsystem B in the upper half in both plots. This would make it easier for the reader to compare the two, and also with Fig. 4 (A).

3. I wondered about the sources [2-6] the authors cite at the beginning of their introduction. For quantum information and entanglement, there are more standard references such as the books by Nielsen & Chuang and by Wilde and also the reviews by Plenio et al. and Gühne & Toth (especially the latter two are highly relevant for this topic), respectively.

4. I suggest the authors to change 'spinful' to 'spin-like' at the end of the discussion.

5. I believe that Figs. B and C are a bit hard to read when printed. Although I guess the authors are aware of this, they may want to reconsider this issue.

RESPONSE TO REVIEWER 1

We are pleased to learn that Reviewer 1 agrees that the “ability to measure entanglement entropy (EE) in condensed matter systems”, as demonstrated in our work, “would have important implications on experiments testing proposals for the realization of exotic electronic states”.

In the report, Reviewer 1 raises four questions, denoted below as $A1$ to $A4$ for convenience. We combine these questions into two groups ($A1+A2$ and $A3+A4$), as the questions in each of these groups form “entangled” pairs by their themes (novelty vs. Bell’s non-locality).

A1. What experiment or analysis could the authors perform that was not possible before, and how would the answer be important?

We thank Reviewer 1 for this crucial question. We believe that our work stands at the frontline in fusing two fundamental topics: mesoscopic transport and quantum information. Indeed, based on our knowledge, our work is the first work that explicitly probes experimentally the entanglement entropy and statistics-induced entanglement via the measurement of current correlations. While the measurements of current correlations performed in our work are similar to those in previous experiments, ***the analysis of entanglement based on the transport measurements was not possible before***. It is the theory developed in the present work that allowed us to connect the two seemingly distinct themes and to extract the information on statistics-induced entanglement from the current noise. The significance of this achievement is emphasized in the first sentence of the Reviewer’s report. Furthermore, the fact that we arrive at statistics-dependent results opens the gate for detecting and extracting statistics-induced entanglement in anyonic systems. Finally, our invented quantities—the entanglement pointer and the statistics-induced entanglement entropy—are in connection to the test of Bell inequality, as we mention in the manuscript and explain in more detail below. Our proposed concepts thus provide an alternative option to probe entanglement, instead of directly testing the Bell inequalities, where much more elaborately designed experiments are required.

A2. How EP and SEE are related to, or distinct from, previous measurements of exchange statistics in quantum Hall “collider” setups such as this one

This is indeed a very natural question, as both our invented functions (EP and SEE) and the measurement of exchange statistics in quantum Hall “collider” setups disclose statistics of quasiparticles. However, there are at least three crucial differences.

Firstly, in comparison to that in quantum Hall “collider” setups, in our work statistics influences the measurement in different ways. More specifically, in quantum Hall “collider” setups [see, e.g., Science **368**, 173–177 (2020) by H. Bartolomei *et al.*], the quasiparticle exchange statistics is probed through “braiding” of non-equilibrium anyons and excitations of anyon-hole pair at the central QPC. Indeed, it is further stated [by e.g., Lee and Sim, Nat. Comms. **13**, 6660 (2022)] that in such an anyonic collider system, contributions from anyonic collisions are less important than that from nonlocal “braiding” (frequently referred to as “virtual braiding” in the literature. In great contrast, in our

setting, statistics is explored by analyzing the effect of “real” two-particle collisions at the central QPC. It is worth emphasizing that the nontrivial “braiding”, which dominates in anyonic systems, does not even exist in fermionic and bosonic systems considered in our work. Importantly, with our collision approach, the information about quasiparticle statistics is extracted directly from processes of braiding two real quasiparticles (supplied by the two terminals), similar to those exchange processes that conventionally illustrate the quantum statistics in textbooks.

As for the second difference, the focus in the research direction of the works mentioned by the Reviewer and the focus of our work are disparate. In the former works, the major topic is to prove the existence of anyonic statistics. Consequently, these experiments do not require multiple sources. Indeed, in, e.g., June-Young M. Lee *et al.*, *Nature* **617**, 277 (2023), the major analysis involves only a single source. In such setups, the very question about the statistics-induced entanglement cannot be asked. Our work, on the contrary, is devoted to measuring entanglement between signals from two sources. Crucially, only with this double-source structure, one can bipartite the system into two subsystems whose quantum states/coherence can be manipulated nonlocally, even if the two subsystems are spatially separated.

To summarize, previous experiments on quantum Hall colliders focused on the demonstration of non-trivial anyonic statistics but did not even consider entanglement (needless to say, those works did not pose questions about statistics-induced entanglement). Entanglement and its quantification are the major topics of our work. Conceptually, both EP and SEE invented in our work pioneeringly bridge two essential quantum mechanical concepts: statistics and entanglement. This connection was never addressed in previous experiments on quantum Hall colliders.

A3. How the metrics EP and SEE relate to entanglement as I perceive it from, e.g., thinking about Bell’s inequalities and two-particle spin or polarization states

We thank Reviewer 1 for asking this very important question, which shows that our discussion of the relation of our entanglement metrics to Bell’s inequalities (presented mainly in Supplemental Information) was not sufficiently clear in the original version of the manuscript (see also comments by Reviewer 2 and our replies to those comments). In short, in a quantum coherent system, our proposed EP and SEE, when close to the theoretically predicted values, reflect the capability of the system to prove Bell non-locality.

As was shown in our manuscript (cf. Fig. 4), the two-particle state after scattering at the central QPC, $|\Psi\rangle$, can be decomposed into a linear combination of $|\Psi_C\rangle$ [which contains states of configurations (2, 0) and (0, 2) with 2 particles in one subsystem and no particle in the other], and $|\Psi_Q\rangle$ [which contains states of configurations (1, 1) with one particle per subsystem]. States of the former one $|\Psi_C\rangle$ resemble the so-called NOON state, and cannot be used for testing Bell’s inequality in the present setup (without tunnel-coupling it to additional external channels). The states $|\Psi_C\rangle$ do not influence cross-correlations between operators from different subsystems. Indeed, after decomposing the

state $|\Psi\rangle$ into $|\Psi_C\rangle$ and $|\Psi_Q\rangle$, the number-operator correlation in this state becomes

$$\begin{aligned}\tilde{S}_{\alpha\beta}(\tau) &\equiv \langle N_\alpha(\tau)N_\beta(\tau) \rangle = \langle \Psi | N_\alpha(\tau)N_\beta(\tau) | \Psi \rangle = (\langle \Psi_C | + \langle \Psi_Q |) N_\alpha(\tau)N_\beta(\tau) (|\Psi_C\rangle + |\Psi_Q\rangle) \\ &= \langle \Psi_C | N_\alpha(\tau)N_\beta(\tau) | \Psi_C \rangle + \langle \Psi_Q | N_\alpha(\tau)N_\beta(\tau) | \Psi_Q \rangle + \langle \Psi_Q | N_\alpha(\tau)N_\beta(\tau) | \Psi_C \rangle + \langle \Psi_C | N_\alpha(\tau)N_\beta(\tau) | \Psi_Q \rangle,\end{aligned}\tag{R1}$$

where N_α refers to the number operator in channel α , which is related to the current operator. Since states of $|\Psi_C\rangle$ have different particle numbers from those in $|\Psi_Q\rangle$, the last two terms above are zero, $\langle \Psi_Q | N_\alpha N_\beta | \Psi_C \rangle + \langle \Psi_C | N_\alpha N_\beta | \Psi_Q \rangle = 0$. Now we move to consider $\langle \Psi_C | N_\alpha N_\beta | \Psi_C \rangle$. Generally, this term is nonzero. However, it vanishes when considering correlations where α and β are from different subsystems. Indeed, $|\Psi_C\rangle$ refers to states of the $(N_A = 2, N_B = 0)$ or $(N_A = 0, N_B = 2)$ configurations, where either subsystem A or B has no particles. These correlations (with α and β labeling drains of different subsystems) are those involved in the test of Bell's inequality. As a consequence, all correlation functions involved in Bell's inequality contain only contribution from $|\Psi_Q\rangle$, i.e.,

$$\tilde{S}_{\alpha\beta} = \langle \Psi_Q | N_\alpha(\tau)N_\beta(\tau) | \Psi_Q \rangle \quad \text{for } \alpha \neq \beta.\tag{R2}$$

Finally, as was shown in Section S3 of Supplementary Information,

$$|\Psi_Q\rangle = \alpha |\uparrow_A\rangle |\uparrow_B\rangle + \beta |\downarrow_A\rangle |\downarrow_B\rangle,\tag{R3}$$

which is an entangled Bell-like state that can be used to test the Bell inequality. Here, the up and down states are orthogonal nonlocal combinations of states in the drain arms of the corresponding subsystems. These states play the role of pseudospin states, as was explained in the manuscript around Fig. 4 and in Section S3 of Supplemental Information. These pseudospin states can be used like real spins in Einstein-Podolsky-Rosen (EPR) steering, see our correspondence with Reviewer 2 below. For pure quantum states, this is equivalent to Bell's non-locality. Thus, the measurement of the EP and SEE can indeed convey messages concerning Bell's non-locality. In addition, measurements of EP and SEE are simpler in comparison to tests of Bell's inequality, where complicated manipulations with the states in the subsystems are required.

In the resubmitted version, we have extended this discussion around Fig. 4 and in Section S3 of Supplemental Information (note that, following the recommendation of Reviewer 2, we have changed the notation: $|\Psi_Q\rangle \rightarrow |\psi\rangle$ and $|\Psi_C\rangle \rightarrow |\tilde{\psi}\rangle$).

A4. How current noise measurements at \mathcal{D}_A and \mathcal{D}_B implement quantum measurements of the type that would be required to test Bell's inequalities.

In our response to A3, we have described the general relation between the test of Bell's inequality and the measured quantities EP and SEE. Importantly, the very structure of Bell's inequalities suggests using the current noise measurements to probe Bell's non-locality. However, a direct test of Bell's inequalities requires a much more involved setup [cf. Phys. Rev. B **66**, 161320 (2002) quoted in Supplemental Information], as additional quantum manipulations on the states of subsystems are necessary. Furthermore, a direct test of Bell's inequality would involve more current noise

measurements, including $\langle I_{\tilde{\mathcal{D}}_A} I_{\tilde{\mathcal{D}}_B} \rangle$, $\langle I_{\mathcal{D}_A} I_{\tilde{\mathcal{D}}_B} \rangle$, $\langle I_{\tilde{\mathcal{D}}_A} I_{\mathcal{D}_B} \rangle$, and $\langle I_{\mathcal{D}_A} I_{\mathcal{D}_B} \rangle$. Here I_α refers to the current operator for the drain α . The approach developed in our work allows one to explore entanglement in the simplest HOM geometry without such additional complications,

RESPONSE TO REVIEWER 2

In the report, Reviewer 2 first summarizes the major conclusion of our work and then asks seven main questions and suggests sixteen modifications. Below we respond to these questions and suggestions in the order they appear in the report. For clarity, seven questions are labeled as $B1$ to $B7$ and comments/suggestions are labeled as $C1$ to $C16$.

Summary: The authors investigate the entanglement content of (quasi-)particles in two spatial regions separated by a quantum point contact (QPC) with respect to their quantum statistics, theoretically as well as experimentally. Their considered setup comprises a two-dimensional electron gas on top of which two diluted electron beams interfere via an electronic analog of the Hong-Ou-Mandel (HOM) interferometer. The entanglement generated by the QPC geometry is accessed in terms of two observables, that is, the entanglement pointer (EP) and the statistics-induced entanglement entropy (SEE), which are designed to capture the quantum statistical contribution to the entanglement of the underlying quasi-particles by subtracting single-source contributions. For instance, the SEE differs by an overall sign for bosons and fermions, see Fig. 3. How this approach is related to Bell non-locality, how to incorporate the effects of additional Coulomb interactions and how the two entanglement quantifiers can be extracted experimentally in the limit of strong dilution is thoroughly discussed. Additionally, the authors realize their proposal and provide evidence for the utility of their methods by comparing how much the interacting and non-interacting curves differ for the cross-correlation function with respect to the EP as well as the SEE. Detailed calculations, additional experimental evidence and an outlook for other applications of their methods is discussed in a supplementary file.

We sincerely thank Reviewer 2 for carefully reading our manuscript and for clearly summarizing the key achievements of our work. We are glad to see that Reviewer 2 appreciated the novel and important results of our work: (i) introduction of two measurable observables, EP and SEE, as quantifiers of statistics-induced entanglement, which fuse statistics and entanglement—two fundamental concepts in quantum mechanics; (ii) clarification of the connection of the approach based on transport noise measurements to Bell’s non-locality, (iii) understanding the role of Coulomb interactions in the experiment, and (iv) remarkable theory-experiment agreement.

I consider the main results, i.e. the EP/SEE and their experimental extraction, interesting and significant.

We are pleased to learn that Reviewer 2 finds our work interesting and significant. We believe that with these conceptual new findings (as confirmed by Reviewer 2), which are supported by experimental measurements, our work more than meets the publication criteria in a high-impact journal such as Nature Communications.

The originality of the EP/SEE is certainly debatable: as acknowledged by the authors, the EP has been discussed

in Ref. [37]. Further, the overall idea of subtracting a certain contribution one wants to get rid of, which underpins both observables of interest, is rather straightforward. Nevertheless, the finding that the SEE becomes directly accessible in the limit of strong dilution is strong as it is backed theoretically and experimentally beyond the non-interacting case, i.e. with Coulomb interactions being present. With their method, the authors solve a crucial problem in the area of quantum transport: determining the quantum statistical contribution to the entanglement of quasi-particles. While this alone may not merit publication in a high-impact journal such as Nature Communications, it is definitely a solid basis for it to be considered.

Concerning the originality of the EP, we emphasize that, while related manipulations with the current correlation functions have been described in Ref. [37] (Ref. [38] of the latest version), the significance of the resulting objects in the context of entanglement has not been understood or appreciated. Based on our knowledge, our work is the first work that explicitly shows how to experimentally obtain statistics-induced entanglement entropy via measurements of current correlations. Our proposed concepts thus provide an alternative option to explore entanglement, instead of testing Bell's inequality, for which much more sophisticated experiments are required.

Conceptual advances above, as well as the support from experimental data, distinguish our work from Ref. [37] (Ref. [38] of the latest version) and other related works, where the relation between entanglement and exchange statistics with noise was not figured out. We are glad to see that Reviewer 2 confirmed that our work solved a critical problem: determining the quantum-statistical contribution to the entanglement of quasi-particles through quantum transport measurements.

B1. I was very surprised that the authors did neither comment on the systematic nor on the statistical errors in their experimental data analysis for Figs. 4B-E, S5 and S6 (has the experiment actually been repeated for a given V_{bias} , i.e. is there any information regarding statistics?). I consider this information crucial especially for evaluating Figs. D and E in the regime of large V_{bias} . It seems that the interacting curve fits better, but without error bars this statement is weak. More generally, it is always important to state the experimental uncertainty somewhere, otherwise the data points may be overinterpreted and the overall significance may be weakened.

We thank Reviewer 2 for this highly constructive question. In the latest version, we have included error bars in Figs. 4, S5, and S6 (the latter two are Figs. S8 and S9 of the latest version, respectively). Notice that we do not provide error bars for Figs. S9A and S9B (of the latest version), since transmissions \mathcal{T}_A and \mathcal{T}_B (for two diluters) were experimentally measured for only one time: these figures serve only as an illustration of the results that were not used for detailed theory-experiment comparison. The inclusion of error bars does not spoil the agreement between theory and experiment.

B2. The authors nicely show that the SEE is accessible in the limit of strong dilution and compare the approximation underlying these findings to higher dilutions in Fig. 3. However, the experimental curves of the SEE in Fig. 4 where extracted beyond the strong dilution regime and computed by a simple rescaling of the strong diluted curves from Fig. 3, which is explained in a bit more detail in SID. Unfortunately, this method is nowhere justified rigorously,

presumably because the two types of curves are obviously not related by a rescaling (this is clearly evident from Figs. 3B and D). I ask the authors to either give a mathematically rigor justification for why a rescaling is appropriate (and not other simple operations such as an addition) at $\mathcal{T}_A = \mathcal{T}_B = 0.2$ and $\mathcal{T} = 0.53$ or to provide any other substantial argument.

We thank Reviewer 2 for this important question, which allowed us to better describe the data analysis in the resubmitted version. Indeed, strictly speaking, only in the strongly diluted limit (i.e., when \mathcal{T}_A and \mathcal{T}_B are small quantities), can one straightforwardly obtain the SEE from the noise measurement. For larger values of \mathcal{T}_A and \mathcal{T}_B , modifications have to be introduced to obtain S_{SEE} from the measured quantity, denoted as \tilde{S}_{SEE} , as was described in the original version of the manuscript. Apparently, our presentation of the corresponding procedure was not sufficiently clear.

In Fig. R1 [with similar plots included in Sec. S1D], we present the ratio $S_{\text{SEE}}/\tilde{S}_{\text{SEE}}$ of the exact SEE, S_{SEE} [cf. Eq. (S21)], and \tilde{S}_{SEE} that is obtainable from the noise measurements. Following Fig. R1, as stated in the main text, the ratio $S_{\text{SEE}}/\tilde{S}_{\text{SEE}}$ approaches one if dilution is strong (i.e., when \mathcal{T}_A and \mathcal{T}_B are small).

FIG. R1: The ratio of \tilde{S}_{SEE} and S_{SEE} for various strengths of dilution. In panels A, B, and C, we choose different values of $\mathcal{T}_A = \mathcal{T}_B$ (A: $\mathcal{T}_A = \mathcal{T}_B = 0.2$, B: $\mathcal{T}_A = \mathcal{T}_B = 0.1$, and C: $\mathcal{T}_A = \mathcal{T}_B = 0.01$), and plot $S_{\text{SEE}}/\tilde{S}_{\text{SEE}}$ as a function of \mathcal{T} . In panel D, we fix the value $\mathcal{T} = 0.53$ and plot the ratio as a function of $\mathcal{T}_A = \mathcal{T}_B$.

Now we go back to Reviewer's question on the justification of the rescaling performed on \tilde{S}_{SEE} . Basically, Fig. R1

indicates that for given values of \mathcal{T}_A , \mathcal{T}_B , and \mathcal{T} , the ratio $S_{\text{SEE}}/\tilde{S}_{\text{SEE}}$ is uniquely determined. This is true if we fix the value of $\mathcal{T}_A = \mathcal{T}_B$ and change \mathcal{T} (A, B and C), or the other way around (D). Consequently, one can obtain S_{SEE} for larger values of \mathcal{T}_A and \mathcal{T}_B by (i) measuring \tilde{S}_{SEE} experimentally, and (ii) rescaling the obtained result with the theoretically obtained ratio $S_{\text{SEE}}/\tilde{S}_{\text{SEE}}$. The nice theory-experiment agreement between the theoretical S_{SEE} and the rescaled \tilde{S}_{SEE} , shown in Fig. 4 of the main text, further supports the validity of our rescaling. In response to this question of Reviewer 2, we have amended Section S1 of Supplemental Information, where we now present the new figures and also describe the rescaling in analytical terms, as suggested by Reviewer 2.

B3. The manuscript lacks scientific rigor whenever concepts of quantum information are discussed in more detail. First, the important distinction between ‘Bell non-locality’ and ‘entanglement’ is not made (see e.g. PRA 104, 052208 (2021) for a recent study I came across when writing this report). In simple words, Bell non-locality is a particularly strong form of entanglement, and detecting true Bell non-locality (for which witnessing entanglement does not suffice), is a notoriously difficult task (which the authors should also mention in their introduction and find appropriate experimental references for).

The confusion of Bell non-locality (see Brunner et al. for a review, which should definitely be cited somewhere) with entanglement becomes more severe in the subsection ‘Entanglement entropy from statistics’ and the corresponding SM section S3. The authors have chosen to describe the state Ψ_C as ‘classically entangled’, which I consider imprecise and misleading (the classical counterpart to entanglement is simply mixedness, which is rather weak form of correlations). The state Ψ_C does actually correspond to the Hong-Ou-Mandel state, a particular kind of a NOON state, which has been investigated in detail in the literature [to be consistent with the literature on this state the authors should also write $\Psi_C = \psi_{2,0} - \psi_{0,2}$ and define $\psi_{0,2}$ accordingly in the SM, which actually comes with a minus sign by default, see (S63)]. Interestingly, this state is not only entangled, but also non-local in the sense that it allows for EPR steering (the hierarchy of entanglement is: Bell non-local > EPR steering > entanglement > separable), see PRA 94, 042119 (2016). Hence, manipulations on one subsystem can influence the other subsystems, but the quantum correlations are not strong enough for Bell non-locality. In contrast, the state Ψ_Q , which does actually correspond to the Bell state (the authors should also mention that), exhibits Bell non-locality and was the state used in the first study demonstrating the violation of Bell’s inequality.

To summarize, the paragraph below (5) and the corresponding discussion in the SM have to be rewritten, the distinction between ‘quantum’ and ‘classical’ entanglement should be changed to ‘Bell non-locality’ and ‘entanglement’ throughout the manuscript and the states Ψ_C and Ψ_Q should be described in more detail, including references to the literature.

We gratefully appreciate the great effort by Reviewer 2 in enhancing the message conveyed by our work, as well as in improving the scientific rigor of our manuscript. We are especially thankful to the Reviewer for careful explanations of terminologies, in particular, regarding ‘Bell’s non-locality’ and ‘EPR steering’, commonly accepted in the quantum-information community, and for pointing out some relevant references to us. This insightful comment by Reviewer 2

has allowed us to clarify the presentation of several crucial points. At the same time, we respectfully disagree with some of the statements made by the Reviewer here, as we are going to explain below. In particular, we do not confuse “Bell non-locality with entanglement” in our manuscript, in contrast to the Reviewer’s assertion.

We note in passing that, following comments by Reviewer 2, *in the latest version*, we have changed the notation for components of the state $|\Psi\rangle$ from $|\Psi_C\rangle$ and $|\Psi_Q\rangle$ to $|\tilde{\psi}\rangle$ and $|\psi\rangle$, respectively. However, *in this response*, we continue using their previous names, i.e., $|\Psi_C\rangle$ and $|\Psi_Q\rangle$, for consistency and to avoid any possible confusion.

We would like to emphasize that the states we address theoretically in the main text are **pure states**. Based on the seminal work by H. M. Wiseman *et al.*, PRL **98**, 140402 (2007), where the hierarchy pointed out by Reviewer 2 was introduced, **Bell non-locality and EPR steering are equivalent to entanglement, as long as pure states are concerned**. Indeed, it is stated in the PRL work by Wiseman and co-authors that “as EPR and Schrödinger noted, steering may be demonstrated using any pure entangled state, and the same is true of Bell non-locality”. This statement is based on abstract theoretical considerations, see, e.g., N. Gisin, *Bell’s inequality holds for all non-product states*, Phys. Lett. A, **154**, 201 (1991). These considerations were further corroborated, e.g., in the work by S. M. Tan *et al.*, PRL **66**, 252 (1991), which proposed practical protocols to violate the Bell inequality with a single-photon NOON state. This proposal, importantly, has been experimentally realized by B. Hessmo *et al.* in PRL **92**, 180401 (2004). For a NOON state, the test of Bell inequality requires the fusion of additional states [the situation of S. M. Tan *et al.*, PRL **66**, 252 (1991)], which is, however, not within our current interest.

The hierarchy described by Reviewer 2 (i.e., **entanglement < EPR steering < Bell non-locality**), applies to **mixed states**: this is exactly the topic discussed in H. M. Wiseman *et al.*, PRL **98**, 140402 (2007) [see also related earlier discussion of Bell’s non-locality of mixed states in D. Collins *et al.*, PRL **88**, 040404 (2002)]. In the resubmitted version of our manuscript, we refer to this important hierarchy in Section S7 “Possible SEE application: quantifying entanglement of a mixed state”, which is devoted to possible generalizations of our theory to mixed states.

Based on the above general statements, the pure state $|\Psi\rangle = |\Psi_C\rangle + |\Psi_Q\rangle$, being a non-product state, is, in principle, amenable to EPR steering and testing the violation of Bell inequality. Thus, the demonstration of the non-product nature of this state is formally sufficient to claim Bell’s non-locality. A direct test of Bell’s non-locality, however, requires sophisticated modifications of the device. In particular, for state $|\Psi_C\rangle$, if forbidding non-local inter-subsystem manipulations (which should be certainly avoided, as they obfuscate the very notion of non-locality) and the fusion of the subsystem with additional external channels, Bell’s non-locality cannot be tested. This point is shared, in the context of NOON states, by, e.g., S. J. Jones and H. M. Wiseman, PRA **84**, 012110 (2011), which has a comment on S. M. Tan *et al.*, PRL **66**, 252 (1991) and B. Hessmo *et al.*, PRL **92**, 180401 (2004), saying “However, all of these experiments relied on **post-selection**, which can be justified on the basis of the fair-sampling assumption for inefficient photodetection in the experiments using weak LOs”. We added the above references to the main text (Refs. [41]-[46]) above Eq. (6), where we discuss the practical implementation of testing Bell’s inequalities with our state $|\Psi_C\rangle$.

This quote brings us back to the similarity of state $|\Psi_C\rangle$ and the NOON state, pointed out by the Reviewer. Indeed, with the Pauli blocking, our $|\Psi_C\rangle$ state appears to be very similar to the bosonic NOON state, as only the total number of particles in the subsystem (2 or 0) matters. We certainly agree with Reviewer 2 that a NOON state actually “is not only entangled but also non-local in the sense that it allows for EPR steering”. This statement (concerning a pure non-product state) definitely follows from the general consideration of entangled pure states discussed above. Indeed, EPR steering on a NOON state was experimentally realized, for instance, in the work by Maria Fuwa *et al.*, Nature Communications **6**, 6665 (2015).

At this point, we would like to emphasize that the state $|\Psi_C\rangle$ **is not exactly equivalent to a NOON state**. Indeed, $|\psi_{2,0}\rangle$ and $|\psi_{0,2}\rangle$ actually correspond to configurations (1, 1, 0, 0) and (0, 0, 1, 1), with each number corresponding to particle numbers in $\tilde{\mathcal{D}}_A$, \mathcal{D}_A , \mathcal{D}_B , and $\tilde{\mathcal{D}}_B$, respectively. This state is, in general, different from a NOON state, where all N composing particles belong to the same mode. This difference can be explored in more sophisticated setups than ours. Specifically, since the particles involved in $|\Psi_C\rangle$ are in different channels (Pauli’s principle), a finite phase difference between these particles can, in principle, be introduced, via, e.g., designing arms of different lengths within one subsystem. As a consequence, one can introduce a phase difference between the components of $|\Psi_C\rangle$, which will not be determined by the number of particles in the subsystem. This is in contrast to the NOON state, where the φ -phase shifter introduces the phase factor that accumulates the phases of all N involved bosons (one cannot manipulate their phases separately):

$$|N\rangle_A|0\rangle_B + |0\rangle_A|N\rangle_B \rightarrow |N\rangle_A|0\rangle_B + \exp(i\varphi N)|0\rangle_A|N\rangle_B.$$

This relative phase of the two components in $|\Psi_C\rangle$ state and in the NOON state can be probed in interference experiments, giving different results. This difference between the NOON state and our $|\Psi_C\rangle$ state is, however, inessential for the discussion of entanglement for the $|\Psi_C\rangle$ state. We thus agree that the state $|\Psi_C\rangle = |\psi_{2,0}\rangle + |\psi_{0,2}\rangle$ in our work is a genuinely entangled state and thank the Reviewer for bringing this analogy to us. In the resubmitted version, we have mentioned the similarity of our $|\Psi_C\rangle$ state to the NOON state in the paragraph discussing Bell’s non-locality of $|\Psi_C\rangle$ above Eq. (6).

At the same time, $|\Psi_C\rangle$ is more rigid than our “Bell-like” state $|\Psi_Q\rangle$, for which extra QPCs placed within the subsystem after the central QPC would do the job. However, such manipulations (which still require much more sophisticated experimental settings than the HOM collider used in our work) are not necessary to quantify the statistics-induced entanglement if utilizing the approach developed in our manuscript. In fact, it is this comparative rigidity of $|\Psi_C\rangle$ with respect to $|\Psi_Q\rangle$ that motivated us to term this state “classically entangled” in the original version of the manuscript. The reason is that, without modifying the present setup, this entangled state cannot be distinguished from the truly classical mixed state of “red and blue shoes” (please see also our response to question A3 of Reviewer 1). Manipulations making the fermions (quantum particles) “green” are forbidden by the Pauli principle, similar to the case of classical “shoes” that cannot become green. However, we agree with the Reviewer that this terminology is misleading, as $|\Psi_C\rangle$ is a truly quantum entangled pure state (see above), so that its “quantumness” can, in principle,

be certified by testing Bell's non-locality, once any possible apparatus is available within the subsystem, including tunnel-coupled external channels. Below Eq. (S66) of the latest Supplementary Information, we have added several paragraphs to discuss connections and differences between $|\Psi_C\rangle$ and a NOON state, as well as $|\Psi_Q\rangle$ and a Bell pair. A comparison of the contributions of states $|\Psi_C\rangle$ and $|\Psi_Q\rangle$ to EP and SEE is now presented in the new Section S4 of Supplementray Information.

Before ending our response to *B3*, we would like to thank Reviewer 2 once again for pointing out these standard concepts/terminologies in the quantum-information community and for questioning some statements from the original version. Following the Reviewer's recommendation, we have modified paragraphs below Eq. (5) of the main text and corresponding discussions in Supplemental Information, to reflect the issues raised by the Reviewer.

B4. Based on the latter considerations, I assume that SEE is actually a quantifier for Bell non-locality. If this is true, this would be a very strong result and I encourage the authors to investigate this deeper. In any case, while I find the comparison with the Bell pairs nice and instructive, I'm not convinced that the SEE does indeed only capture Bell non-locality. It might well be that it also evaluates to a non-zero real number if the state under considerations exhibits EPR steering. Hence, I ask the authors to provide a mathematical rigor argument for why SEE is zero if and only if the state exhibits Bell non-locality. If such a statement can not be provided, a few statement in the paragraph below Eq. (6) have to be weakened.

Also, I ask the authors to clarify whether the SEE is a true entanglement measure, an entanglement monotone, or an entanglement witness (the term 'entanglement quantifier' used by the authors is rarely used and usually means entanglement monotone, but I have also seen it as a synonym for entanglement witness), see also my comment 5 below.

Out of curiosity: the definition of the SEE, ie. Eq. (4), looks very similar to the definition of the mutual information. I'm not sure, but may the two are related. I encourage the authors to briefly investigate their relations, but this is optional. My motivation is the following: if there is a simple relation, it may become obvious why the SEE does only capture Bell non-locality.

To begin with, we emphasize again that the state we are addressing is a pure state, where there is no difference between the two classes of quantum correlations, "Bell non-locality" and "EPR steering", mentioned by Reviewer 2 (see our response above to *B3*).

Regarding the "mathematical rigor argument for why SEE is zero if and only if the state exhibits Bell non-locality", we believe that Reviewer 2 means *non-zero*. In fact, in the original version of the manuscript, in saying "hence measurement of pseudospins in a 'transverse direction', as required by Bell's inequalities.... This statistics-induced entanglement is captured by the SEE", we did not imply such a statement (i.e., "if and only if"). Our text only stressed that effective Bell pairs do contribute to SEE. Given the possible confusion, as noticed by Reviewer 2, we rephrased this paragraph, and stress more on the statistics-induced entanglement.

Indeed, in the statistics-irrelevant case, where quantum particles can be considered as distinguishable from each other (the "blue particle" and the "red particle"), one particle can reside in subsystem *A* and the other in subsystem *B*.

One can then form an entangled state (amenable, in principle, to the Bell-inequality test), which carries no statistical information. This state leads to non-vanishing entanglement entropy, yet both the EP and SEE vanish, since the two-particle scattering in this case is equivalent to two independent single-particle processes.

Our EE and SEE can thus be considered as witnesses of statistics. As for entanglement in general, these functions are entanglement measures in the Reviewer's terminology. As for the relation of the SEE to mutual information, the analogy is indeed compelling. However, we do not find any straightforward formal connection between these objects in the present setting. At the same time, this analogy is indeed useful for quantifying entanglement in mixed states, as discussed in Section S7 (see also our response to question B5). In view of this analogy, we have added mutual information to the list of the entanglement measures discussed in the literature in the context of mixed states.

B5. In S6 the authors claim that the SEE is able to quantify the entanglement of globally mixed states. This is usually the field of entanglement measures, of which the authors have mentioned and referenced two prominent ones. Proving that these two are indeed entanglement measures is not straightforward at all, and I ask the authors to support their claim with mathematics or instead weaken their statements.

We thank Reviewer 2 for this very important question. In short, we are not claiming that these two quantities mentioned in the original version, i.e., negativity and topological EE, are valid and widely applicable functions in the quantification of mixed-state entanglement. Instead, these two quantities are chosen as examples of trials and proposals (in the existing literature) that were considered as potential candidates for quantifying the mixed-state entanglement.

More specifically, we would like to emphasize that Sec. S7 (the previous Sec. S6) simply discusses, prospectively, a possible future application of our suggested SEE. In the latest version of Supplementary Information, we have rephrased this sentence, into "To quantify entanglement in mixed states, multiple potential candidates, such as logarithmic entanglement negativity, mutual information, and topological EE (see, e.g., Refs. [S18, S19]) have been discussed in the literature."

Further, after this sentence, we write to refer to the effort that studies entanglement in mixed states via the hierarchy discussed in the Reviewer's question B3 above: "In addition, the hierarchy of quantum correlations, including the EPR steering and Bell nonlocality were introduced, to classify entanglement of mixed states: entanglement < EPR steering < Bell nonlocality (see, e.g., Ref. [S20], and related discussions of Refs. [S21, S22]. Denoted as hierarchy, it means that Bell nonlocality of a mixed state is sufficient to perform EPR steering (and to prove entanglement), but not the other way around (i.e., not all entangled mixed state can realize EPR steering; not every EPR steerable mixed state is subject to Bell's nonlocality inequality)."

B6. Out of interest: the theoretical discussion comprises both fermions and bosons, while the experimental discussion does (of course) not. Is there an easy way of investigating the entanglement of bosons with a similar setup, e.g. by considering Cooper-pairs?

Experimental studies in our work involve only a fermionic system realized with a quantum Hall setup that provides (weak-interaction) quantum-coherent chiral channels for fermions. Motivated by our findings, we anticipate exper-

imental investigations (perhaps, by other groups) of the entanglement of bosons in, e.g., a photonic cavity system, where particles are typically well-controllable on large coherence distances.

We agree with Reviewer 2 that a Cooper-pair system is, in principle, capable of realizing the proposed experiment with bosons. However, performing such experiments with a Cooper-pair setting appears to be more challenging, because of several potential problems, e.g., difficulty in realizing chiral 1D platforms that carry Cooper pairs and the possible decoherence between two (would-be) entangled fermionic composites of a Cooper pair. Importantly, when two such composite particles (i.e., four fermions) scatter at the same QPC, additional scattering channels are possible, such as the destruction of bosons into individual fermions and an “exchange” between the two bosons by their constituents. These interesting complications are not included in our theoretical consideration that addressed statistics-induced entanglement of “elementary” particles. Therefore, we consider photonic platforms as better candidates to verify our theory on the bosonic side. Generalizing our theory to describe entanglement of composite particles is an important challenge for future work (as mentioned in the outlook of our work in connection to composite fermions exhibiting anyonic statistics; see also our response to question B/ below).

B7. Also out of interest: in the discussion the authors state that their method can be generalized to anyons. However, the measurable SEE relies on the entanglement pointer, which itself does only distinguish bosons/fermions by the overall sign and hence, there is no room left. Can you provide any intuition for why your ansatz does not break down for anyons?

We thank Reviewer 2 for this important question. Briefly, the fermionic and bosonic situations are the two extremes. For anyonic systems, the SEE and EP will “interpolate” between the fermionic and bosonic cases. This can be seen within the two-particle scattering picture. Indeed, for two particles with statistical angle $\pi\nu$ (where $\nu = 0$ for bosons, and $\nu = 1$ for fermions), the anti-bunching probability equals [see, e.g., PRL **109**, 106802 (2012) by G. Campagnano *et al.*] $(1 - \cos \pi\nu)/2$, where the second term, i.e., $-\cos \pi\nu/2$ reflects the statistics. Following the qualitative consideration in our manuscript, the SEE of anyons is expected to be proportional to this anti-bunching probability. With this dependence on the statistical angle, for the most widely studied Laughlin state with $\nu = 1/3$ (for which the anti-bunching probability is reduced compared to distinguishable particles), the SEE will be negative, the same as the bosonic case. The SEE amplitude will be, however, smaller than that of a bosonic system (where $\nu = 0$).

Now we move to sixteen comments/suggestions from Reviewer 2, C1 to C16.

C1. The definition of the entanglement entropy in the introduction is wrong. It lacks a crucial overall minus sign [the definition in the SM is correct in (S10), but also wrong above (61)]. Maybe this also implies that the overall minus sign in Eq. (4) should be dropped, but this is anyway mere convention.

We thank Reviewer 2 for pointing out this typo. It has been corrected in the latest version.

C2. The usage of bra’s and ket’s to denote state vectors is inconsistent throughout the manuscript. For instance,

in the paragraph below Eq. (6) the state vectors up/down are written without ket's. Or, in (S58), the authors have equated a real number with a state vector in Hilbert space.

We thank Reviewer 2 for pointing out this inconsistency. In the latest version, all states are written within the symbol “ $|\rangle$ ”.

C3. Similarly, the usage of operator hats is inconsistent. The current operators are equipped with operator hats, while all other operators are not. Here, a single clarifying sentence should suffice.

We thank Reviewer 2 for meticulously reading our manuscript and for this constructive suggestion. In the latest version, we have removed hat for all operators, to avoid this inconsistency.

C4. In the information theory literature, it is rather uncommon to denote the composition of two subsystems (or random variables) A and B by $A + B$. The latter usually refers to the sum of the two, while their composition is denoted by AB . Note also that the entropy of the sum is a lower bound on the entropy of the composition. Further, measures of correlations between A and B are usually written with $A : B$, see e.g. in the caption of Fig. S6.

We thank Reviewer 2 for this suggestion. We decided to replace $A + B$ with “the combined Hilbert space $A \oplus B$ ” when referring to the total system in the Introduction. We have also replaced S_{AB} in the caption of Fig. S9 (of the latest version) with $\langle I_A I_B \rangle - \langle I_A \rangle \langle I_B \rangle$. We have made this modification to avoid possible confusion from other quantities e.g., entropy and SEE, where the letter S is also used. As a result, we do not need to use the (suggested by the Reviewer) notation $A : B$ for correlation functions involving operators from subsystems A and B .

C5. It is ‘von Neumann’ and not ‘Von Neumann’, see e.g. the sentence above (S61).

We thank Reviewer 2 for pointing out this typo. It has been corrected in the latest version.

C6. What the authors call ‘irreducible correlation’ is usually referred to as ‘connected correlation function’ and written with a subindex c .

We thank Reviewer 2 for this comment. In the latest version, we have added the term “connected correlation function”, when we first refer to irreducible correlators [after Eq. (1) of the main text]. We have also replaced “correlations” with “correlators” when introducing the subscript “ir” there. However, we would like to continue using this subscript referring to the term “irreducible correlator” in our work, as it is a term that is conventionally used in the condensed-matter community. Indeed, in this community, the term “irreducible correlator” or “irreducible correlation function”, which refers to correlators after removing the product of averages of each individual operator, is used to distinguish the “connected correlation function” from the “reducible” correlators that contain information from products of averages.

C7. Some abbreviations are not defined, are defined after their first usage or are defined multiple times. For example, ‘SEM’ is not defined, ‘2DEG’ is defined too late, and ‘HOM’ is defined twice..

We thank Reviewer 2 for the extremely careful reading of our manuscript. We have included definitions of these abbreviations.

C8. There is maybe an obsolete space bar in the last down ket w.r.t. B in the caption of Fig. 2.

After carefully checking this issue, We did not find any “obsolete space bar” in the source TeX file.

C9. In the majority of the manuscript, the authors use “”, but sometimes they instead use ‘’ (see e.g. ‘collisions’ or ‘generating’ function).

We thank Reviewer 2 for pointing out this stylistical inconsistency in our manuscript. In the latest version, we write all the quotation marks consistently, by changing e.g., ‘collisions’ and ‘generating’ into “collisions” and “generating”.

C10. Below Eq. (6) the authors write ‘ “pseudospin” of subsystem A’. Please elaborate briefly what is meant here.

We thank Reviewer 2 for this constructive suggestion. Briefly, with one particle in each subsystem, there are two orthonormal eigenstates corresponding to “one electron in subsystem A ”, and two orthonormal eigenstates corresponding to “one electron in subsystem B ”. These “pseudospin” eigenstates are referred to as $(|A \uparrow\rangle, |A \downarrow\rangle)$ and $(|B \uparrow\rangle, |B \downarrow\rangle)$, respectively, cf. Eq. (S67) of the Supplementary Materials. We have added a sentence after Eq. (6) of the main text, to explain the meaning of “pseudospins” (please also see our response to question A3 by Reviewer 1).

C11. I recommend the authors to simplify the comparison of Figs. S5 and S6 to the main Fig. 4 by showing the quantities of interesting as functions of V_{bias} (if there is no good reason against it).

We thank Reviewer 2 for this constructive suggestion. Figures S5 and S6 are Figs. S8 and S9 of the latest version, where we have changed the label of the x -axis to bias voltage V_{bias} .

C12. The authors have chosen to denote the system’s total state by ρ_0 . Given that ρ_1 to ρ_4 label eigenvalues of the reduced density matrix, see e.g. (S61), I suggest to use ρ or ρ_{AB} instead.

Equation (S61) corresponds to Eq. (S64) of the latest Supplementary Information. We agree with Reviewer 2, that in view of the notation $\rho_1 - \rho_4$ introduced in Eq. (S64), ρ_0 of the main text may appear confusing to readers. We have thus followed the suggestion by the Reviewer and replaced ρ_0 of the main text with ρ_{AB} .

C13. I personally would find it more convenient to have Figs. 1(A) and (B) in the same order, that is, having subsystem B in the upper half in both plots. This would make it easier for the reader to compare the two, and also with Fig. 4(A).

We thank Reviewer 2 for this useful suggestion. Following this suggestion, we have modified Figs. 1A and 4A to match other schematics.

C14. I wondered about the sources [2-6] the authors cite at the beginning of their introduction. For quantum information and entanglement, there are more standard references such as the books by Nielsen & Chuang and by Wilde and also the reviews by Plenio et al. and Gühne & Toth (especially the latter two are highly relevant for this topic), respectively.

We thank Reviewer 2 for the suggested references. We have replaced the references used in the previous version with those suggested by the Reviewer.

C15. I suggest the authors to change ‘spinful’ to ‘spin-like’ at the end of the discussion.

Here, by “spinful” we are indeed referring to real spin. In our work, the system contains particles that are spin-

polarized in a strong magnetic field (Zeeman effect): only one spin projection is involved. Such spin-polarized electrons can be equivalently considered as “spinless” fermions. We however anticipate that our theory can be generalized to systems where particles do have the spin degree of freedom – i.e., to the case of truly spinful particles.

C16. I believe that Figs. B and C are a bit hard to read when printed. Although I guess the authors are aware of this, they may want to reconsider this issue.

We guess that Reviewer 2 was pointing to Figs. 4B and 4C. In the latest version, we have rearranged sub-figures of Fig. 4 to achieve better readability.

I. LIST OF CHANGES

- In the Introduction, we have modified the references when referring to quantum information processing.
- We have replaced “Von” with “von” in the Introduction and the Supplementary Information [before Eq. (S63)].
- We have corrected the typo (the missing of a minus sign) in the definition of entanglement entropy in the Introduction and Supplementary Information.
- In the second paragraph of the Introduction, we denote the total density matrix as ρ_{AB} (instead of ρ_0) and refer to the space where it acts as “in the combined Hilbert space $A \oplus B$ ” (instead of referring to the entire system as “ $A + B$ ”).
- We have added the definition of SEM in the caption of Fig. 4. We have moved the definition of 2DEG to the paragraph before the section “The model and the entanglement pointer (EP)”. We keep only the definition of HOM in the Introduction.
- We have modified Figs. 1A and 4A, such that now the upper side of both figures represents subsystem A , following the arrangement in other schematics, e.g., Figs. 1B and 2.
- We have removed all hats for current operators, in both the main text and Supplementary Information.
- We have added the term “connected correlation function” when we first define the subscript “irr” referring to irreducible correlators, after Eq. (1).
- In the resubmitted manuscript, we denote $|\Psi_C\rangle$ and $|\Psi_Q\rangle$ of the previous version as $|\tilde{\psi}\rangle$ and $|\psi\rangle$, respectively.
- We have refined discussions on $|\tilde{\psi}\rangle$ and $|\psi\rangle$, including their connections to Bell non-locality after Eq. (5) of the main text, with relative references provided. Corresponding changes are also introduced in the caption of Fig. 2.
- After Eq. (6) of the main text, we have refined discussions on the definition and rotation of pseudospin. We have also explained how the statistics-dependence of SEE emerges from the expressions for $|\psi\rangle$ and $|\tilde{\psi}\rangle$.
- We have included error bars in Fig. 4 of the main text and Figs. S8 and S9 of Supplementary Information.
- We have rearranged Fig. 4 of the main text to improve its appearance in the printed format.
- At the end of Sec. S1C, we have added several lines of discussion and a plot (Fig. S1 of the latest version), addressing the analytical justification of the rescaling of \tilde{S}_{SEE} .
- We have included a new plot (Fig. S2) in Sec. S1D, when discussing rescaling of the SEE.

- In Fig. S6 of the latest version, we have added two panels, Figs. S6F and S6G (with x -axis the bias), to provide a more direct comparison to Fig. 4 of the main text.
- Below Eq. (S66), we have added several lines to discuss the difference/similarity between $|\psi\rangle$ and a Bell state, as well as $|\tilde{\psi}\rangle$ and a NOON state.
- We have added a new section, Sec. S4, to discuss individual contributions of $|\psi\rangle$ and $|\tilde{\psi}\rangle$ to SEE.
- In Sec. S7 of the latest version (Sec. S6 of the previous version), we have refined our statement on the efforts of the quantum community to quantify entanglement of mixed states. These efforts include proposed entanglement quantification functions, and the classification of mixed states into different entanglement hierarchies.
- Before “Summary and outlook”, we have added several lines, to compare our statistics-induced entanglement with that by interactions.

Reviewer #2 (Remarks to the Author):

Dear authors and editors,

with even more interest I have read the revised manuscript "Measuring statistics-induced entanglement entropy with a Hong-Ou-Mandel interferometer" and the author's response to my feedback. First, let me stress that I am very happy about the amount of effort the authors have put into answering all my questions (I was well aware of my report being demanding for the authors). To not further exhaust all the people being involved, I promise to keep things simple this time (I only address open questions below, everything else was handled very well).

Response to the authors

B1. It is reassuring to see that the error bars do not spoil any of the important conclusions. However, I could not find any statement on what the error bars are representing. Please indicate how the error bars were calculated, i.e. whether they represent 1 or 2 standard deviations and how often the experiment has been repeated.

B3. I am impressed by the huge efforts made by the authors to improve the quantum information parts of their manuscript. I agree with almost all statements, changes, and improvements. But first, I apologize for being imprecise about the non-locality hierarchy when considering pure states (in particular, I wrongfully stated that the NOON state cannot reveal Bell non-locality, which is of course nonsense). However, the reason why I insisted so much on distinguishing Bell's non-locality from entanglement is rooted in the simple fact that preparing a pure state in an experiment is close to impossible as a result of noise. Therefore, in all serious works concerned with experimental demonstrations of Bell non-locality, criteria valid for mixed state were used. Of course, from a theoretical perspective, the output state of the HOM interferometer is pure, and no such distinction has to be made. I ask the authors to reflect on this issue and then decide on whether they want to include a clarifying half sentence around Eq. (5). Besides, the clarifications regarding the NOON state - in the main text as well as in the supplement - are very much appreciated.

B5. The statements regarding the EP and SEE have been improved, but unfortunately, the statements regarding entanglement measures are now less precise than before (it seems like we have misunderstood each other here). To clarify: The logarithmic negativity is an entanglement measure (actually the one which is easiest to compute), the mutual information is the measure for all correlations (classical and quantum) and thus only useful if the state is pure in which case it reduces to two times the entanglement entropy, and the topological entanglement entropy measures the topological order, which is a certain contribution to the entanglement entropy appearing in quantum many-body systems. Hence, please only refer to the logarithmic negativity as an entanglement measure (and please also do not state that this is a potential candidate as it is indeed a true measure) and delete the other two. A reference to the review by the Horodecki family (where also many other measures are discussed) should suffice at this point.

C4. The combined Hilbert space is a product space, and hence the direct sum should be replaced by a tensor product.

Finale recommendation

Given the satisfying answers not only to my but also to the comments of referee I as well as the minor nature of the remaining requests, I now strongly recommend the manuscript for publication in Nature

Communications after a last minor revision. I thank the authors again for their stamina in answering my questions and wish them the best in their future endeavors.

Reviewer #3 (Remarks to the Author):

Entanglement is an important but elusive property of quantum mechanical systems, a nonlocal correlation that is easy to define formally but difficult to extract from a measurement. The simplest test for entanglement, the Bell inequality, has been widely studied. The present manuscript introduces more informative measures of the degree of entanglement and shows how they can be measured in a semiconductor nanostructure. This is an innovative approach in the electronic context. I find the manuscript well presented and convincing. I have read the previous two referee reports, which raise valid questions that the authors have answered in a satisfactory way, in my opinion. My recommendation is to accept this for Nature Communications.

RESPONSE TO REVIEWER 2

We greatly thank Reviewer 2 again for showing interest in our work. We are pleased to learn that finale verdict by Reviewer 2 is to “strongly recommend the manuscript for publication in Nature Communications after a last minor revision”. We also sincerely appreciate Reviewer 2 for the great effort in helping us to improve our manuscript.

Concerning our previous response, Reviewer 2 raises four questions/suggestions below, in the latest referee report. Below we respond to each of them separately.

B1. It is reassuring to see that the error bars do not spoil any of the important conclusions. However, I could not find any statement on what the error bars are representing. Please indicate how the error bars were calculated, i.e. whether they represent 1 or 2 standard deviations and how often the experiment has been repeated.

We thank Reviewer 2 for this question. In the latest version, we introduce the meaning of error bars in the caption of Fig. 4 and Fig. S8 (1σ standard deviation of the mean for the set of measurements). In addition, we explain how we obtained the error bars, with a concrete defining equation, in Sec. S6A.

B3. I am impressed by the huge efforts made by the authors to improve the quantum information parts of their manuscript. I agree with almost all statements, changes, and improvements. But first, I apologize for being imprecise about the non-locality hierarchy when considering pure states (in particular, I wrongfully stated that the NOON state cannot reveal Bell non-locality, which is of course nonsense). However, the reason why I insisted so much on distinguishing Bell's non-locality from entanglement is rooted in the simple fact that preparing a pure state in an experiment is close to impossible as a result of noise. Therefore, in all serious works concerned with experimental demonstrations of Bell non-locality, criteria valid for mixed state were used. Of course, from a theoretical perspective, the output state of the HOM interferometer is pure, and no such distinction has to be made. I ask the authors to reflect on this issue and then decide on whether they want to include a clarifying half sentence around Eq. (5). Besides, the clarifications regarding the NOON state - in the main text as well as in the supplement - are very much appreciated.

We thank Reviewer 2 for the clarification. We also fully agree with Reviewer 2 that for mixed states, where experiments are mostly focused on, the proof of Bell nonlocality is more demanding than that of quantum entanglement. In the latest version, this fact is stressed at the end of Page 3 of the main text in the paragraph below Eq. (6). In particular, we give there a reference to Sec. S7 of Supporting Information, where entanglement in mixed states was already addressed.

B5. The statements regarding the EP and SEE have been improved, but unfortunately, the statements regarding entanglement measures are now less precise than before (it seems like we have misunderstood each other here). To clarify: The logarithmic negativity is an entanglement measure (actually the one which is easiest to compute), the mutual information is the measure for all correlations (classical and quantum) and thus only useful if the state is pure in which case it reduces to two times the entanglement entropy, and the topological entanglement entropy measures the topological order, which is a certain contribution to the entanglement entropy appearing in quantum many-body

systems. Hence, please only refer to the logarithmic negativity as an entanglement measure (and please also do not state that this is a potential candidate as it is indeed a true measure) and delete the other two. A reference to the review by the Horodecki family (where also many other measures are discussed) should suffice at this point.

Following comments from Reviewer 2, we have modified the text in Sec. S7 of SI.

C4. The combined Hilbert space is a product space, and hence the direct sum should be replaced by a tensor product.

We thank Reviewer 2 for pointing it out. We have replaced $A \oplus B$ by $\mathcal{H}_A \otimes \mathcal{H}_B$, explicitly referring to Hilbert spaces.

I. LIST OF CHANGES

- In Introduction, we have replaced $A \oplus B$ by $\mathcal{H}_A \otimes \mathcal{H}_B$, when referring to the Hilbert space of the system.
- At the end of Page Three, we have added several lines, to state the fact that in mixed states, the probing of Bell nonlocality is more demanding than the prove of quantum entanglement.
- We have added the meaning of error bars in the captions of Figs. 4 and S8. How to measure the error bars are further explained in Sec. S6A.
- We have modified the text of Sec. S7, when discussing the entanglement quantifier of a mixed state.

Reviewer #2 (Remarks to the Author):

All my questions/suggestions have been answered/implemented satisfactorily. The manuscript is ready for publication in Nature Communications in its present form.